

# Changes in Holocene meridional circulation and poleward Atlantic flow: the Bay of Biscay as a nodal point

Mary, Yannick (1), Eynaud, Frédérique (1), Colin, Christophe (2), Rossignol, Linda (1), Brocheray, Sandra (1, 3), Mojtahid, Meryem (4), Garcia, Jennifer (4), Peral, Marion, (1, 5), Howa, Hélène (4), Zaragosi, Sébastien (1), Cremer, Michel (1)

*(1) Laboratoire Environnements et Paléoenvironnements Océaniques et Continentaux (EPOC) -UMR 5805, Université de Bordeaux, 33615 Pessac, France*
*(2) Laboratoire Géosciences - Université de Paris-Sud, 91405 Orsay Cedex, France*
*(3) now at: Institut Polytechnique LaSalle-Beauvais – Dpt Géosciences, 19 rue Pierre Waguet – BP 30313 – 60026 Beauvais, France*
*(4) UMR CNRS6112 LPG-BIAF, Recent and Fossil Bio-Indicators, Angers University, 2 Bd Lavoisier, 49045 Angers CEDEX 01, France*
*(5) now at Laboratoire des Sciences du Climat et de l'Environnement (LSCE-IPSL),  Domaine du CNRS, bât.12 - 91198 Gif-sur-Yvette, France*

*Keywords*

Meridional circulation, Bay of Biscay, Holocene, Sea surface temperature, North Atlantic, Subpolar and Subtropical Gyres

*Abstract*

This paper documents the last 10 ka evolution of one of the key parameters of climate: sea-surface temperatures (SST) in the subpolar North Atlantic. We focus on the southern Bay of Biscay, a highly sensitive oceanographic area because of its strategic and nodal position regarding the dynamics of the North Atlantic subpolar and subtropical gyres. This site furthermore offers unique sedimentary environments characterized by exceptional accumulation rates, enabling the study of Holocene archives at (infra)centennial scales. Our results mainly derive from planktonic foraminiferal association analysis on two cores from the southern Landes plateau. These associations were used as quantitative tools (thanks to the Modern Analog Technique) to track past hydrographical changes. SST reconstructions were thus obtained at an unprecedented resolution and compared to a compilation of Holocene records from the northern North Atlantic. From this regional perspective are shown fundamental timing differences between the gyre dynamics, nuancing classical views of a simple meridional overturning cell.



## 1. Introduction


At climatic and shorter meteorological scales, the key role of the North Atlantic oceanic
circulation in climate changes is no longer debatable (e.g., Clark et al., 2002; Bryden et al., 2005). The
Atlantic Meridional Overturning Circulation (AMOC) and its dynamics are critical regarding the
amplitude and frequency of climate modulations over Europe (westerlies, droughts and/or stormy periods,
e.g. Dawson et al., 2007; Magny et al., 2003; Sorrel et al., 2009; Trouet et al., 2012, Van Vliet-Lanoe et
al., 2014a and b, Jackson et al., 2015). The two related North Atlantic gyres, the subpolar gyre (SPG) and
the subtropical gyre (STG) are fundamental for these processes as they transfer heat and salt toward the
Nordic seas (e.g., McCartney and Mauritzen, 2001; Perez-Brunius et al., 2004; Hatun et al., 2005) where
convection occurs (e.g. Lozier and Stewart, 2008). Their expansions and contractions notably control the
inflow from the North Atlantic Current (NAC) to higher latitudes, thus also affecting the heat budget of
the Greenland-Iceland-Norwegian seas, which is critical in the meridional climatic balance (i.e., Hatun et
al., 2005, Thornalley et al., 2009). However, complex feedbacks force nonlinear responses within the
Earth's radiative budget, preventing climate sciences from providing a precise view of which processes
are at play; the incapacity of models to "correctly" represent the last decade of instrumental data is one of
the strongest illustrations of this appraisal (e.g., Ba et al., 2014; Karl et al., 2015; Fyfe et al., 2016).
During the late Holocene, STG and SPG latitudinal and/or longitudinal migrations contributed to well-
known climatic anomalies in Western Europe, such as the Little Ice Age or the Medieval Warm
Period/Anomaly, and probably played a major role at longer time scales (Copard et al., 2012; Sorrel et al.,
2012; Staines-Urias et al., 2013). By providing the first Holocene inventory of (infra)centennial
hydrographic changes in the inner Bay of Biscay, this paper aims at testing Western European temperate
oceanic signals *vs.* those from a broader North Atlantic view with a focus on the SPG dynamics, this latter
being seen as a key component of the AMOC variability (e.g. Hatun et al., 2005; Thornalley et al., 2009;
Colin et al., 2010). Our study site (Figure 1) is ideally located under the temperate eastern limb of the
NAC, in the southern Bay of Biscay and close to STG/SPG divergence zone. This geographic
configuration provides to this marine environment a high sensitivity regarding Northern hemisphere
climatic signals at present (e.g. Le Cann and Serpette, 2009; Esnaola et al., 2012; Garcia-Soto and



Pingree, 2012) with some sedimentary archives furthermore evidencing a strong potential to track down
the Holocene variability (Mojtahid et al., 2013; Garcia et al., 2013; Brocheray et al., 2014; Mary et al.,

2015).

Today, the Bay of Biscay is characterized by a complex, variable sea-surface circulation with

strong seasonal changes, marked by a September-October versus March-April - *SOMA* pattern (Pingree
and Lecann, 1990; Pingree and Garcia-Soto, 2014). The main surface current in the Bay of Biscay is the
European Slope Current (ESC), flowing northward along the Armorican Shelf (Figure 1), with important
spatial and seasonal variations (Garcia-Soto and Pingree, 2012; Charria et al., 2013). Circulation can
reverse during summer along the shelf break, flowing weakly southwestward (Charria et al., 2013). In
autumn-winter, the northward flow reaches a maximum, especially when combining with southern
intrusions from the Iberian Poleward Current (IPC) which flows along the western Iberian margin (e.g.
Peliz et al., 2005) before turning eastward at the Cape Finisterre (NW Spain). The IPC northward
extension into the Bay of Biscay is known as the Navidad Current (e.g. Garcia-Soto et al., 2002; Le Cann
and Serpette, 2009). The winter compound of IPC and ESC is designated as the European Poleward
Current (EPC, Garcia-Soto and Pingree, 2012), and drives relatively warm and saline water to the Nordic
seas, contributing to their heat and salt budget. The Bay of Biscay is additionally strongly marked by
surface water inflow coming from the North Atlantic Current (Figure 1), which enters the Bay from its
northwestern boundary (Pingree, 2005; Pingree and Garcia-Soto, 2014; Ollitrault and Colin de Verdiere,
2014). In contrast with surface circulation of the inner Bay of Biscay, the NAC water inflow shows only
limited seasonal variability. At inter-annual time scales however, NAC oscillations are mainly driven by
westerly wind regime (Pingree, 2005), and consequently by the North Atlantic Oscillation (NAO), one of
the key modes of climatic oscillation in the North Atlantic. So far, little is known about long term
oscillations of the NAC inflow into the Bay. Modern surveys of SST variability over the last 150 years in
the Bay of Biscay report that temperature oscillations are mainly controlled by the Atlantic Multi-decadal
Oscillation (AMO, Garcia-Soto and Pingree, 2012). The influence of the NAO on SST in the Bay of
Biscay is more complex and contributes only little to the observed long term trend, although sharp, inter-
annual changes of the NAO index impact annual SST variability (Garcia-Soto and Pingree, 2012).



Moreover, NAO conditions influence large-scale oceanic circulation patterns indirectly responsible for
surface temperature anomalies over the Bay (Pingree, 2005; Garcia-Soto and Pingree, 2012).

The present paper is based on analyses conducted on two high-resolution well dated cores from

the southern part of the inner Bay of Biscay (Figure 1, Table 1): core KS10b (e.g. Mojtahid et al., 2013)
and core PP10-07 (e.g. Brocheray et al., 2014). These cores show exceptionally high sedimentation rates
for the Holocene, up to 200 cm.ka$^{-1}$ for core PP10-07, and 86 cm ka$^{-1}$ for core KS10b. Here we present
reconstructed SST data derived from an ecological transfer function based on the Modern Analogue
Technique (see Methods) applied to planktonic foraminifera assemblages. These Bay of Biscay sea-
surface reconstructions are compared to selected North Atlantic Holocene records onwards a data mining
exercise done in the frame of the French ANR HAMOC (Holocene North Atlantic Gyres and
Mediterranean Overturning dynamic through Climate Changes) project database (see http://hamoc-
interne.epoc.u-bordeaux1.fr/doku.php?id=start) and referencing sea-surface reconstructions of high time-
resolution.

## 2. Methods

### 2.1. Age models

Updated age models have been built for the Bay of Biscay cores. All raw $^{14}$C ages were calibrated

and converted to calendar ages using the Marine13 calibration curve and the recommended age reservoir
of 405 years (Reimer et al., 2013), as no adequate and robust local age reservoir values exist in the area
(see Mary et al., 2015 for a discussion). Smooth-spline regression based on the published $^{14}$C dates (n =12
for core Ks10b, Mojtahid et al., 2013) were applied (Figure 2). For core PP10-07, two supplementary $^{14}$C
dates were obtained at the top of the sequence (Table 2) and the age model was built using a 5 degree
polynomial regression (Figure 2). Core MD03-2693 age model (also exploited in this paper) was built
using linear interpolation based on published $^{14}$C and $^{210}$Pb (n=3 and n=8, respectively, Mary et al., 2015).
Age-depth modeling and calibration were performed using the dedicated software Clam (Blaauw, 2010),
written in the open-source statistical environment R (http://www.r-project.org/).




## 2.2. Past hydrographical parameter quantification

Planktonic foraminifera (PF) assemblages were used to quantify sea-surface parameters: species

abundances were determined (counts of 300 specimens at least) on the > 150 µm fraction from
sedimentary aliquots retrieved at maximum 10 centimeter-intervals along the studied cores, thus giving a
mean time resolution of 50 and 150 years for core PP10-07 and KS10b respectively (see Supplementary
material for detailed data). SST reconstructions were calculated using the Modern analog technique
(MAT) a method successfully developed on PF (e.g, Pflaumann et al., 1996; Kucera et al., 2005; Telford
and Birks, 2011; Guiot and de Vernal, 2007; 2011). The calculations derive from modern spectra
previously compiled and tested separately in the frame of the MARGO exercise for the North Atlantic
Ocean and the Mediterranean Sea respectively (Kucera et al., 2005; Hayes et al., 2005). They are based
on sediment surface samples analyzed for their contents in PF (specific relative abundances) and thus
offer the advantage of already having integrated regional taphonomic processes. At EPOC
(Environnements et Paléoenvironnements Océaniques et Continentaux) laboratory, these two MARGO
databases were summed to provide larger analog choices and ambiguous data points were excluded (i.e.
undated points showing anomalies in the biogeographical distribution), resulting in a final training set of
n=1007 modern analogs. Modern sea-surface parameters were extracted from the WOA ATLAS with the
sample tool developed by Schäfer-Neth and Manschke (2002). The latter was developed for the MARGO
program and interpolates the 10 m World Ocean Atlas WOA -1998 mean seasonal and mean annual
temperatures over the four existing data points surrounding the sample location (see http://www.geo.uni-
bremen.de/geomod/staff/csn/woasample.html) thus providing spatio-temporal averaged values of SST
(see Kucera et al., 2005 for MARGO analytical developments and Telford and Kucera, 2013 for further
considerations).
Calculations were run under the R software with the BIOINDIC package (ReconstMAT script) developed
by J. Guiot (https://www.eccorev.fr/ spip.php?article389) using relative abundances of PF with no





mathematical transformation (no logarithmic or square root transformations which are frequently used to
increase the equitability within assemblages for instance).
Past hydrological parameter values are derived from a weighted average of the SST values of the five best
analogs. The maximum weight is given for the closest analog in terms of statistical distance (i.e.
dissimilarity minimum). The ReconstMAT script furthermore includes the calculation of a threshold
regarding this statistical distance which prevents calculation in the case of poor- or no- analogous
situations. The degree of confidence of this method allows reconstructing seasonal and annual SST with a
maximum root mean square error of prediction (RMSEP) of 1.3°C (see Supplementary material). This
method (named MATR_1007PF for Modern Analog Technique derived from 1007 modern spectra of PF
assemblages) was extensively tested at EPOC including comparisons with similar MAT developed
regionally on PF (e.g. Salgueiro et al., 2008; 2010) providing very coherent reconstructions along the
western European margin (see Eynaud et al., 2013 for details) and producing pertinent
paleoceanographical data (see Penaud et al., 2011; Sánchez Goñi et al., 2012; Sánchez Goñi et al., 2013
for records also produced with MATR_1007PF).

**3. Holocene SST oscillations in the Bay of Biscay**
Despite the different bathymetric and physiographic positions of the studied cores (Figure 1, Table
1), reconstructed annual SST in the Bay of Biscay show coherent oscillations of remarkably similar
timing (Figure 3a). Small amplitude differences are observed between the two focused records, but
synchronous warm periods are clearly identified between 8.2-7.4 ka BP and 6.6-5.6 ka BP, these intervals
roughly corresponding to the upper and lower limits of the mid-Holocene hypsithermal in the North
Atlantic region (e.g. Eynaud et al., 2004; Walker et al., 2012; Tanner et al., 2015).
On historical time-scales, warm intervals are detected in both cores between 2.6-1.8 ka BP
(Roman Warm Period, RWP) and 1.2-0.5 ka BP (Medieval Warm Period, MWP), although less obvious
in core KS10b because of the lower time resolution. An offset of up to 4°C above mean annual modern
values is observed during a large temperature excursion around ca 2 Ka BP in core PP10-07 only. The
amplitude of the warmings detected between 8.2-7.4 ka BP and 6.6-5.6 ka BP reaches concomitantly 2 to





3°C in both records. Such amplitudes in the detected SST warm pulses are especially high in comparison
to modern annual values. However, considering the strong modern seasonal SST variations in the Bay of
Biscay (as shown on Figure 3a), a 4°C shift of mean annual SST is coherent with a deviation of annual
mean temperature toward mean summer values.

Comparison of the southern Bay of Biscay SST reconstructions with other records from the

Western European margin (Figure 3 and 4) suggests that the observed millennial-scaled warm episodes
are coherent features which reflect typical climatic patterns, at least expressed regionally, but also
probably more broadly. Indeed, further along the Bay of Biscay margin, other high resolution Holocene
archives reveal similar and synchronous episodes. Concomitantly to the observed warm SST pulses also
seen within the seasonal means (see Supplementary material), Holocene pollen assemblages from core
VK03-58 bis (Naughton et al., 2007) indicate a decrease in mean annual precipitations; this drought being
related, according to the authors, to a change in the seasonality with warmer summers especially. In the
same way, the evolution of coccolithophorid concentrations in the subpolar North-Atlantic along the
Irminger Current pathway, interpreted as indicating stronger contribution of NAC water toward the
Nordic seas (Andrews and Giraudeau, 2003; Giraudeau et al., 2004, Moros et al., 2012), showed strong
similarities with the Bay of Biscay SST signals. Peaks in coccolithophorid abundances in cores B997-330
and MD99-2269 (Figure 4e and f) (see location on Figure 1) were recorded synchronously to the warm
pulses in the Bay of Biscay, with especially positively marked anomalies detected around 2 ka BP and 8
ka BP. The Bay of Biscay SST oscillations further correspond with those reconstructed from marine
records from the Barents Shelf (see location on Figure 1) from core MSM5/5-712-2 (Werner et al., 2013,
Figure 3c) and core M23258 (Sarnthein et al., 2003; Figure 3d). This coherency suggests teleconnections
between the southern Bay of Biscay and the Nordic seas, probably due to a common driving mechanism
linked to the NAC inflow vigor and to the modulation of its split off Ireland between the SPG and the
STG.

In between the observed warm intervals, SST reconstructions of core PP10-07 and KS10b reveal

several low values slightly colder than today (Figure 3a). The time interval between 5.6 and 2.6 ka BP is
characterized by temperatures around -1°C compared to the modern ones. This period roughly



corresponds to the late Holocene Neoglacial Cooling (e.g. Eynaud et al., 2004; Wanner et al., 2008;
Walker et al., 2012). In the same way, short-lived events of 2°C cooling are visible around ca 8.2, 7, 4,
2.9 and 1.7 ka BP (Figure 3 and 4). The two older anomalies are synchronous and well-marked in both
KS10b and PP10-07 cores.

The comparison of the timing of these cold spells to other existing Holocene reconstructions from

the North Atlantic Ocean reveals that they represent coherent and reproducible features (Figure 4).
Interestingly, density anomalies thought to reflect millennial-scale variability in the SPG dynamics
(Thornalley et al., 2009; Farmer et al., 2011) were synchronously recorded at sub-thermocline depths in
the southern Iceland basin. These anomalies were interpreted as reflecting a strong /*weak*, longitudinally
extended/*contracted* SPG thus driving more/*less* vigorous but fresher/*saltier* Atlantic inflow throughout
the Faroe current branch and thus modulating the AMOC strength (Thornalley et al., 2009). The good
temporal correspondence between the cold spells detected in core PP10-07 (even if shorter) and the
density anomalies (core RAPiD-12-1K, Figure 4h) registered in the subpolar North-Atlantic support, as
seen for warm events, a direct teleconnection with the inner Bay of Biscay, probably throughout a
STG/SPG seesaw which would influence tracks/intensities of the temperate westerlies. The short lived
cold anomalies of PP10-07 are furthermore concomitant with periods of increased storminess identified in
various coastal sediments from the NW European margin (Holocene Storm Periods after Sorrel et al.,
2012, Figure 3g). These periods have been related to a weakened, westward contracted SPG, involving a
rapid feedback in the atmospheric dynamics.
In the subtropical North Atlantic, study of benthic foraminiferal stable isotopes in core EUGC-3B
(located in the Galician Shelf, Pena et al., 2010; see Figure 1) also showed similar cold anomalies which
were interpreted by the authors as suggesting enhanced contribution of colder, NE Atlantic ENACW
waters reaching the Iberian margin during these events.





## 4. The European poleward current and the influence of subtropical sourced waters in the Bay of Biscay

Modern surveys (e.g. Garcia-Soto et al., 2002; Lozier and Stewart, 2008; Garcia-Soto and Pingree, 2012) and paleoceanographic time-series (e.g., Mojtahid et al., 2013) recently evidenced the influence of the IPC, and its extension in the Bay of Biscay (i.e. Navidad Current, Garcia-Soto et al., 2002), on surface circulation and hydrological conditions along the European Margin. At present, these incursions of warm waters in the bay occur during winter under specific seasonal wind regimes (of southerly wind off Portugal and westerly wind off Northern Spain, Charria et al., 2013) and negative anomalies of sea level pressure over the North Atlantic (Pingree and Garcia-Soto, 2014). While these conditions were previously related to a negative mode of the NAO (Garcia-Soto et al., 2002), recent analysis of instrumental time-series showed that weather conditions responsible for Navidad current may not always correspond to a fixed value of the NAO index (Pingree and Garcia-Soto, 2014). The Navidad Current occasionally creates warm SST anomalies, enhanced transport of warm water through the pole and could thus be the vector of planktonic exotic (from subtropical origin) faunal invasions in the inner Bay of Biscay (see Mojtahid et al., 2013 and Garcia et al., 2013 for example in the fossil record; see Garcia-Soto and Pingree, 2012 and Pingree and Garcia-Soto, 2014 for example in instrumental time-series) which could bias our SST reconstructions. In the following, we thus examine the hypothesis of a persistent poleward surface current during the Holocene that would have triggered the observed SST warm anomalies in the PP10-07 and KS10b records.

In order to test the coherency of surface hydrographic features along the temperate and subtropical adjacent portions of the European margin, we compared Bay of Biscay SST reconstructions with existing SST (annual) records produced along the Iberian Margin (Figure 3b and c). We first test this link over historical times, compiling SST high resolution data obtained on the proximal core MD03-2693 (after Mary et al., 2015), which matches accurately those from core PP10-07 (see Figure 3a between 0.5 and 1.5 ka and also Figure E4 in the Supplementary material), with additional high resolution records (Figure 3b). The combination of these records reveals a slight warming associated to the Medieval Warm Period and coherent low-amplitude multi-decadal SST oscillations which echoes those of AMO anomalies as reconstructed by Mann et al (2009). Especially striking is the high degree of synchronicity detected between the Iberian



margin (core PO287-06 , Abrantes et al., 2011) and the Bay of Biscay at the scale of the last 1.5 ka, despite
differences in the proxies used to generate paleo-SST (Alkenones vs MAT on PF respectively) and age-
model uncertainties (which probably explain offsets of a few hundred years around 1200 A.D.). The good
coherency with AMO reconstructions further supports modern oceanographic assumptions of AMO driving
multi-decadal change of SST in the area (Garcia-Soto and Pingree, 2012) and shows that this modulation is
at least valid for the late Holocene. Interestingly, modern winter incursions of Iberian water through the Bay
of Biscay take place during periods of increasing AMO (Garcia-Soto and Pingree, 2012). During these
episodes, warm winter anomalies of up to 1.1° are observed in the Bay of Biscay, which are consistent with
the amplitude of the warmings detected in both MD03-2693 and PP10-07 past reconstructions.

However, at a longest Holocene perspective, existing SST records from the Iberian margin do not

reveal any coherent patterns with those from the Bay of Biscay over the last 10 ka (Figure 3C).
Regardless of the proxies involved in SST reconstructions (Alkenones and MAT), there is no evidence of
any earlier distinct SST excursions in the high time resolution data of the Iberian cores MD99-2331,
D13882, MD95-2042 and MD01-2444 (Figure 3c, see also Figure E4 in the Supplementary material) or
elsewhere in other lower resolution Holocene records from the same area (Naughton et al., 2007; Martrat
et al., 2007; Rodrigues et al., 2009; Voelker and de Abreu, 2011; Chabaud et al., 2014). The early
Holocene SST reconstructions in this area show a monotonous long term decrease of SST correlated with
the Holocene decline of summer insolation (e.g. Marchal et al., 2002, see also Figure E4) which contrasts
strongly with the warm episodes observed in core PP10-07 and KS10b at that time (see Figure E4 in the
Supplementary material). Taking into account the similarities between late Holocene records in the
Iberian margin and in the Bay of Biscay, our data thus suggests a disconnection between these two
regions during the first part of the Holocene, up to 1.5 ka BP. We interpret this divergence as a distinct
response of the Bay of Biscay to North Atlantic millennial changes in the NAC/SPF system dynamics
(e.g. Perez-Brunius et al., 2004) whereas southwestern Europe has probably undergone a mixed influence
of diverse subtropical climatic trends. Sea-surface environments from the Bay of Biscay, located at the
interface between the SPG and STG influences may have, as currently observed in frontal regions,
recorded an amplified signature of NAC shifts, themselves driven by contraction/extension phases of the





whole North-Atlantic gyre system (STP, SPG, and Polar Gyre also). To decipher the role of each of these
gyres is at present not possible on the basis of our records only, and requires additional high-resolution
comparable marine archives along a latitudinal gradient at least between 30° and 60°N. The analyses of
the influence of Mediterranean hydrographic changes (via the Mediterranean outflow export especially)
together with those linked to the Eastern North Atlantic Upwelling Region would also be very important
to tackle in such a context.

## 5. Implication for Holocene climate dynamics


In agreement with modern climate observations (e.g. Ba et al., 2014), North Atlantic paleoceanographic
studies describe a strong impact of the Subpolar gyre (SPG) dynamics on the NAC inflow toward high-
latitudes and global circulation during the Holocene (Bianchi and Mc Cave, 1999; Oppo et al., 2003;
Perez-Brunius et al., 2004; Thornalley et al., 2009; Giraudeau et al., 2010; Moros et al., 2012; Staines-
Urías, 2013). Freshwater fluxes in the Labrador Sea and wind stress over the North Atlantic are key
drivers of eastern extensions/contractions of the SPG (Hatun et al., 2015), thus also controlling the
salinity balance over the North Atlantic, boreal deep-water convection and North hemisphere climate
patterns. The compilation of proxy-records from further south in the Bay of Biscay indicates that the
Holocene relatively long-term periods of warming in the Bay of Biscay are interbedded/superposed to
rapid, millennial cold anomalies of SPG origin (Figure 4). In agreement with other North Atlantic
records, strong NAC occurs preferentially during the Holocene optimum (Berner et al., 2007; Solignac et
al., 2008), and during the Roman Warm Period (Werner et al., 2012). In contrast, the occurrences of cold
anomalies in the North Atlantic follow a 1500 years periodicity during the Holocene (e.g., Thornalley et
al., 2009; Debret et al., 2007; Sorrel et al., 2012), and are accurately reflected by the SST PP10-07 record
(Figure 4f).
As also suggested by recent studies of modern time-series (Lozier et al., 2010; Lozier, 2012), Holocene
SST records from the Bay of Biscay evidence a decoupling of gyre dynamics, and a potential gyre-
specific expression of the AMOC. Model studies similarly question the meridional coherence of the



AMOC, revealing an inherent character of its mid-latitude variability at decadal time-scales (Bingham et
al., 2007), mainly driven by wind forcing and eddy variability. While our findings support coherent sea-
surface hydrographical patterns between subtropical and temperate environments along the western
European margin, suggesting a coupled SPG/STG gyre dynamics over the last 1.5 to 2 ka, earlier
Holocene contexts seem to have been rather favorable to a gyre-specific expression, as seen at least from
SST reconstructions. To understand climatic processes behind these observations and test their coherency
region per region, a pan-(North)-Atlantic view is required, urging for comprehensive data compilation
efforts as those undertaken for instance in the work conducted for the Ocean2k SST synthesis (e.g.
McGregor et al., 2015). SST records should however been incremented by complementary parameters
when possible, especially to document hydrographic processes at various depth, in order to better
understand the 3D articulation of the oceanic thermal and dynamic responses to various Holocene
forcings (e.g. changes in insolation, sea-level - gateway connection, volcanism, or even anthropogenic
related, which could have been cumulative or not).

## 6. Conclusion

Our study, which documents Holocene surface hydrographical changes at unprecedented time-
scales in the Bay of Biscay, reveals contrasted patterns which accurately reflect the variability of the
North Atlantic gyre dynamics. Coherently with stronger NAC inflow in the Nordics seas as detected in
other archives from the northern North Atlantic, our high-resolution sedimentary records identify specific
warm periods during the early Holocene and at ca. 2 ka BP and reveal that northward advection of
subtropical waters may have influenced SST oscillations in the Bay of Biscay during the last 1.5 ka BP.
In addition, SST signals from the Bay of Biscay show the occurrences of short-term cold anomalies,
interpreted here as the signature of changes in SPG dynamics. The influence of the two main North
Atlantic gyres, i.e STP vs SPG, observed asynchronously over most of the Holocene in the Bay of Biscay,
indicate fundamental differences in the temporal variability of their dynamics, contrasting with the idea of
a coherent, basin-wide-driven, overturning cell in the North-Atlantic. Our results suggest a gyre-specific





expression of the AMOC, which may contribute to strong regionalisms in the response of the North
Atlantic hydrography to Holocene climatic changes.

**7. Acknowledgments**

Analyses documented in this study have been supported by the French ANR HAMOC. We are

grateful to the captain and crew of the RV *Pourquoi Pas?* and to the scientific team of the 2010-
SARGASS cruise. This work beneficiated from $^{14}$C AMS measurement facilities thanks to the ARTEMIS
French project. We thank Giovanni Sgubin, Didier Swingedouw and Eleanor Georgiadis for useful
discussions and comments on the manuscript. This is an UMR EPOC contribution. Data will be set on
http://www.pangaea.de/.

M.Y. and E.F. designed the study and wrote the paper in the frame of the ANR HAMOC project
coordinated by C.C..
E.F., R.L., M.M., G.J., P.M., H.H. performed and/ or supervised planktonic foraminifera assemblage
analyses and picking for the datings. E.F. ran the transfer function. M.Y. performed age modelling with
the help of E.F. and M.M.. B.S, S.Z. and C.M. investigated the sedimentology of core PP10-07. All
authors contributed to discussions and interpretation of the results. The authors declare no competing
financial interests.



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



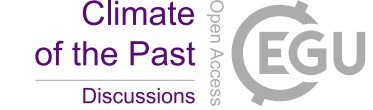

______________________________________________________________
**Table caption**
**Table 1**: Location and references of the southern Bay of Biscay cores used in this study.
**Table 2**: Summary of AMS $^{14}$C ages of core PP10-07 with calendar correspondences.





| Cruise, Core label | Latitude (°N) | Longitude (°E) | Water depth (m) | Longitudinal distance (km) from the shore | Datasources and references |
|---|---|---|---|---|---|
| SARGASS, PP10-07 | 43.677 | -2.228 | 1472 | 58 | **This work,** Brocheray et al., 2014 |
| PROSECAN IV, KS10b | 43.833 | -2.050 | 550 | 50 | **This work,** Mojtahid et al., 2013 |
| SEDICAR/PICABIA, MD03-2693 | 43.654 | -1.663 | 431 | 15 | **This work,** Gaudin et al., 2007, Mary et al., 2015 |

**Table 1:** Location and references of the southern Bay of Biscay cores used in this study.

| Depth in core PP10-07 (cm) | Sample | Material | Ref Number | mg C | d¹³C | pMC corrected | | | RAW 14C Age yr BP | | | corrected reservoir age / -400 yr BP | Calibrated age CLAM yr BP | -2σ yr | +2σ yr | Error yr | Confidence % |
|---|---|---|---|---|---|---|---|---|---|---|---|---|---|---|---|---|---|
| **4,5** | PP10-07, 3-6 cm (TR1) | Bulk planktonic foraminifera | SacA39103 | 0,572 | 0,12 | 90,6 | ± | 0,24 | **790** | ± | 30 | *390* | ***423*** | *353* | *493* | *70* | *92,6* |
| **124,5** | PP10-07 124-125 cm (TR2) | Bulk planktonic foraminifera | SacA39104 | 0,455 | -1,1 | 82,1 | ± | 0,24 | **1590** | ± | 30 | *1190* | ***1149,5*** | *1063* | *1236* | *86,5* | *95* |
| **219,5** | PP10-07 218-221 | Bulk planktonic foraminifera | SacA 29590 | 0,7 | 0,2 | 77,5 | ± | 0,19 | **2050** | ± | 30 | *1650* | ***1618*** | *1533* | *1702* | *85* | *95* |
| **380** | PP10-07 / 380 | Bulk planktonic foraminifera | SacA 26975 | 0,78 | -4,6 | 72,2 | ± | 0,24 | **2615** | ± | 30 | *2215* | ***2271*** | *2175* | *2366* | *96* | *95* |
| **720,5** | PP10-07 / 720-721 | Bulk planktonic foraminifera | SacA 26976 | 1 | -0,9 | 58,8 | ± | 0,22 | **4265** | ± | 30 | *3865* | ***4380*** | *4272* | *4487* | *108* | *95* |
| **1050** | PP10-07 / 1050 | Bulk planktonic foraminifera | SacA 26977 | 1,1 | -5,2 | 49,4 | ± | 0,17 | **5660** | ± | 30 | *5260* | ***6070*** | *5970* | *6170* | *100* | *95* |
| **1180** | PP10-07 1180 | Bulk planktonic foraminifera | SacA 29591 | 0,69 | -0,3 | 44,6 | ± | 0,14 | **6490** | ± | 30 | *6090* | ***7007*** | *6897* | *7116* | *110* | *95* |
| **1540** | PP10-07 / 1537-1543 | Bulk planktonic foraminifera | SacA 26978 | 1,17 | -1,9 | 33,8 | ± | 0,17 | **8705** | ± | 40 | *8305* | ***9371*** | *9276* | *9466* | *95* | *95* |
| **1731,5** | PP10-07 1730-1733 | Bulk planktonic foraminifera | SacA 29592 | 0,84 | -0,8 | 33 | ± | 0,12 | **8900** | ± | 30 | *8500* | ***9556*** | *9477* | *9635* | *79* | *95* |
| **1981,5** | PP10-07 1980-1983 | Bulk planktonic foraminifera | SacA 29593 | 1 | -1,5 | 31,6 | ± | 0,12 | **9270** | ± | 30 | *8870* | ***10093*** | *9992* | *10193* | *101* | *92* |


**Table 2:** Summary of AMS ¹⁴C ages of core PP10-07 with calendar correspondences.



**Figure caption**
**Figure 1:** A) map showing the regional scheme of the main surface currents in the Bay of
Biscay, drawn after the compilation of modern hydrological survey from Pingree and Garcia-
Soto (2014). North Atlantic Current (NAC), Iberian poleward Current (IPC), and European
Slope Current (ESC) are respectively represented by the red and orange arrows. The studied
sedimentary cores PP10-07 and KS10b from the inner Bay of Biscay are shown in red.
Additional Holocene records cited in the text are displayed by green squares. B) North
Atlantic general circulation pattern (SPG: Subpolar Gyre, STG: Subtropical Gyre, EPC:
European Poleward Current, after Lherminier and Thierry, 2015) with the location of the
northern and southern sedimentary records discussed in the text. Core references: **1**-Brocheray
et al., 2014; **2**-Mojtahid et al., 2013; **3**-Gaudin et al., 2006; Mary et al., 2015; **4**-Naughton et
al., 2007a; **5**-Pena et al., 2010; **6**-Werner et al., 2013; **7**-Sarnthein et al., 2003; **8**-Giraudeau et
al., 2004; **9**-Andrews and Giraudeau, 2003; **10**-Thornalley et al., 2009; **11**- Naughton et al.,
2007b; **12**-Abrantes et al., 2011; **13**-Chabaud et al., 2014; **14**- Rodrigues et al., 2009; **15**-
Martrat et al., 2007.

**Figure 2:** Revised age models for cores KS10b, MD03-2693, and PP10-07 (left panels)
compared to previous published age models (right panels with original references).

**Figure 3:** Mean Annual sea surface temperature (SST) records from the Western European
margin. A) Holocene SST signals from cores PP10-07 and KS10b (this study) reconstructed
using the Modern Analog Technique (MAT) based on planktonic foraminifera (see Methods),
and compared to SST signal of the adjacent core MD03-2693 (Mary et al., 2015). Black dots
identify $^{14}$C age control points. B) SST signals spanning the last 1500 years in the Bay of
Biscay (core MD03-2693) based on MAT and from the Iberian Margin (core PO287-06,



Abrantes et al., 2011) using alkenones. Reconstructed signals are compared with the AMO
reconstruction of Mann et al., (2009). The dotted curve represents core MD03-2693 signal
transposed on top of the two other curves. C) Holocene SST signals from the Iberian Margin
using MAT based on planktonic foraminifera for cores MD99-2331 (after Naughton et al.,
2007b) and MD95-2042 (after Chabaud et al., 2014) and Alkenones for cores D13882 (after
Rodrigues et al., 2009) and MD01-2444 (after Martrat et al., 2007).

**Figure 4:** Comparison of annual SST Holocene signals from the Bay of Biscay (A and B)
with records from the northern North Atlantic highlighting variations of the NAC intensity
and SPG dynamics; C) SST signal of core MSM5/5-712-2 (Fram strait, Werner et al., 2013)
and of D) core M23258 (Barents shelf, after Sarnthein et al., 2003), both reconstructed using
the Modern Analog Technique (MAT) based on planktonic foraminifera; E) Concentration of
NAC indicator coccolith species in core MD99-2269 (North Iceland Shelf, after Giraudeau et
al., 2004) and in F) core B997-330 (North Iceland Shelf, after Andrews and Giraudeau, 2003);
The PP10-07 record is here also plotted by a thin dotted red line to underline the comparison;
G) Holocene Storm Periods (after Sorrel et al., 2012) reconstructed from sedimentological
evidence from a compilation of coastal cores in North-western Europe; H) core Rapid-12-1K
(Thornalley et al., 2009) proxy for upper-water column stratification, calculated using derived
Mg/Ca and $\delta^{18}$O temperatures and salinities of *G. bulloides and G. inflata*. Dotted vertical
lines point out events of density anomalies at sub-thermocline depths in the southern Iceland
basin.
Changes in gyre circulation dynamics are compared with the Holocene division of Wanner et
al. (2008). The topmost arrows indicate periods of probable weak SPG also corresponding to
cold anomalies in the southern Bay of Biscay. Pink bands conversely highlight periods of
warmth which also correspond to enhanced NAC activity North of Iceland.



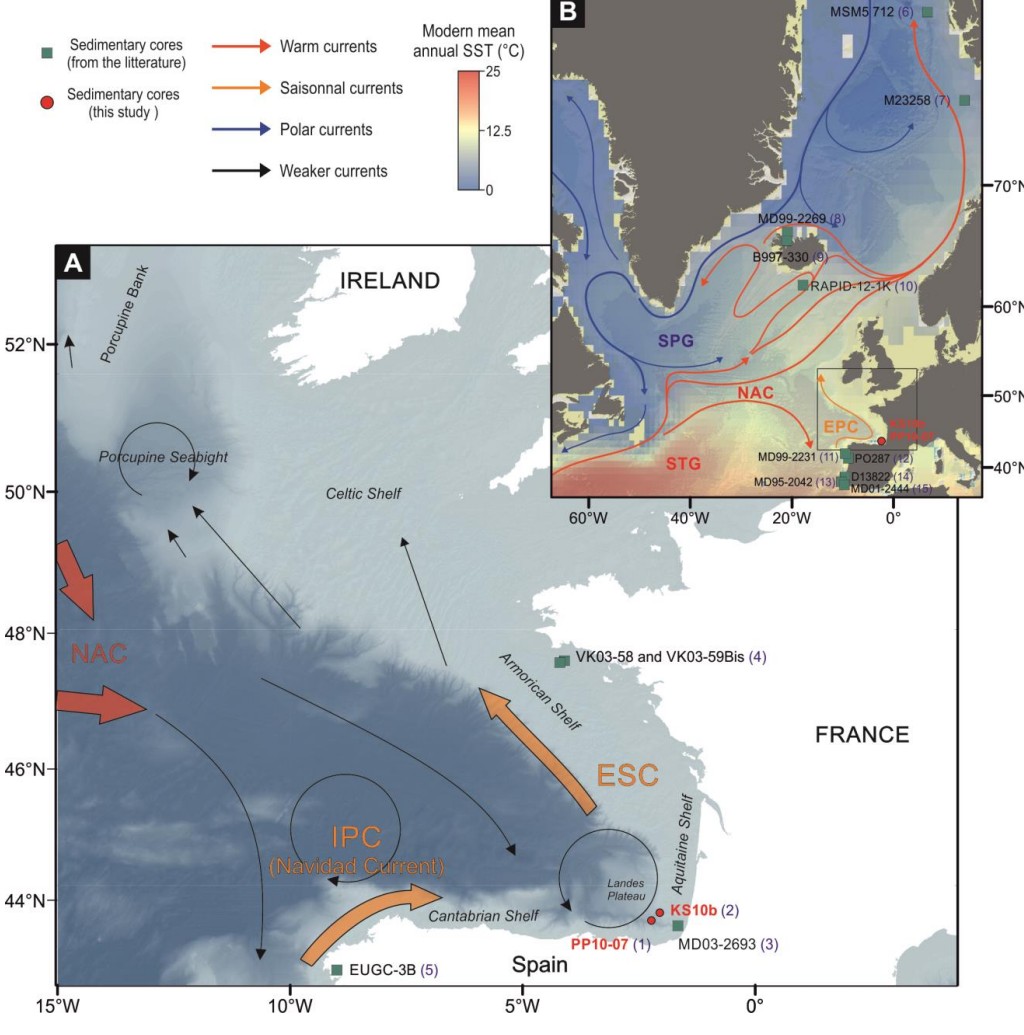


**Figure 1**: A) map showing the regional scheme of the main surface currents in the Bay of Biscay, drawn
after the compilation of modern hydrological survey from Pingree and Garcia-Soto (2014). North Atlantic
Current (NAC), Iberian poleward Current (IPC), and European Slope Current (ESC) are respectively
represented by the red and orange arrows. The studied sedimentary cores PP10-07 and KS10b from the
inner Bay of Biscay are shown in red. Additional Holocene records cited in the text are displayed by green
squares. B) North Atlantic general circulation pattern (SPG: Subpolar Gyre, STG: Subtropical Gyre, EPC:
European Poleward Current, after Lherminier and Thierry, 2015) with the location of the northern and
southern sedimentary records discussed in the text. Core references: 1-Brocheray et al., 2014; 2-Mojtahid
et al., 2013; 3-Gaudin et al., 2006; Mary et al., 2015; 4-Naughton et al., 2007a; 5-Pena et al., 2010; 6-
Werner et al., 2013; 7-Sarnthein et al., 2003; 8-Giraudeau et al., 2004; 9-Andrews and Giraudeau, 2003;
10-Thornalley et al., 2009; 11- Naughton et al., 2007b; 12-Abrantes et al., 2011; 13-Chabaud et al., 2014;
14- Rodrigues et al., 2009; 15- Martrat et al., 2007.



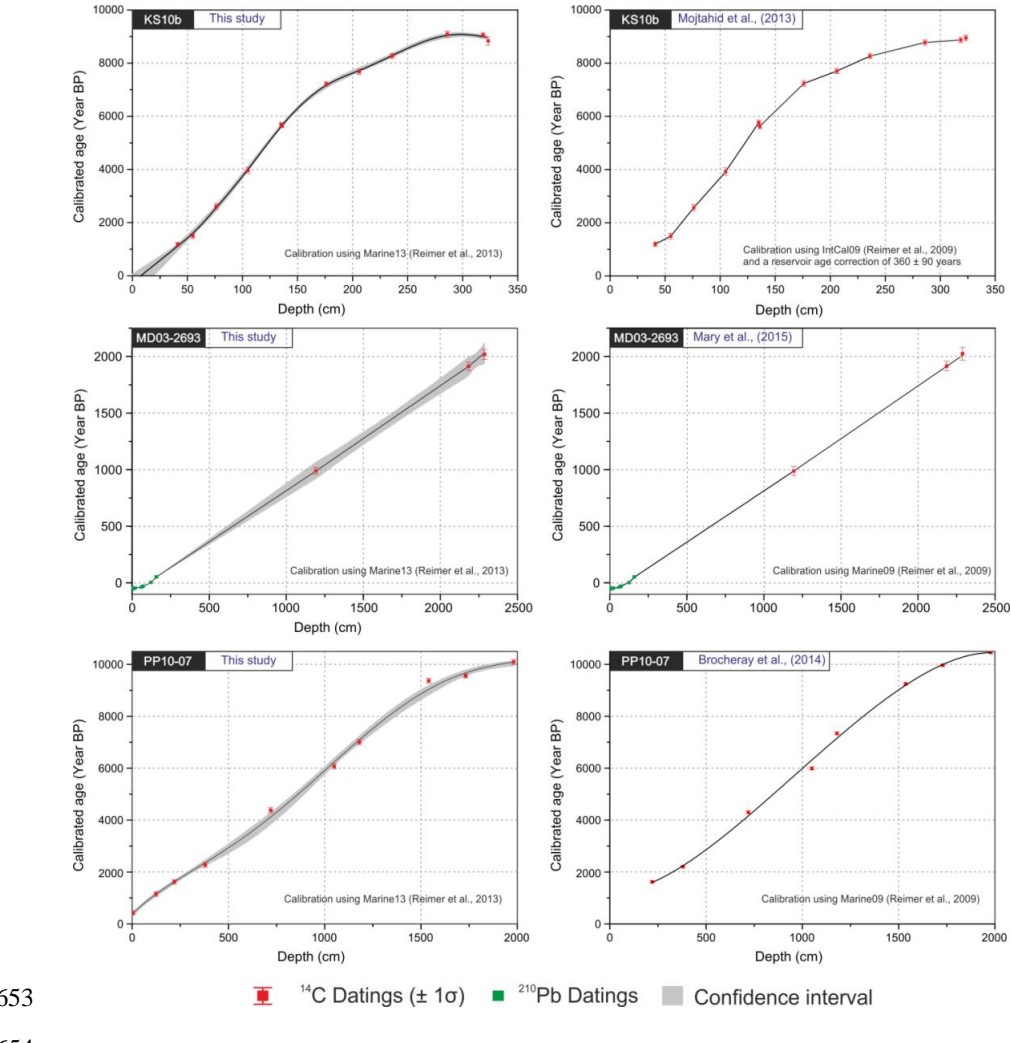

**Figure 2:** Revised age models for cores KS10b, MD03-2693, and PP10-07 (left panels) compared to previous published age models (right panels with original references).



657

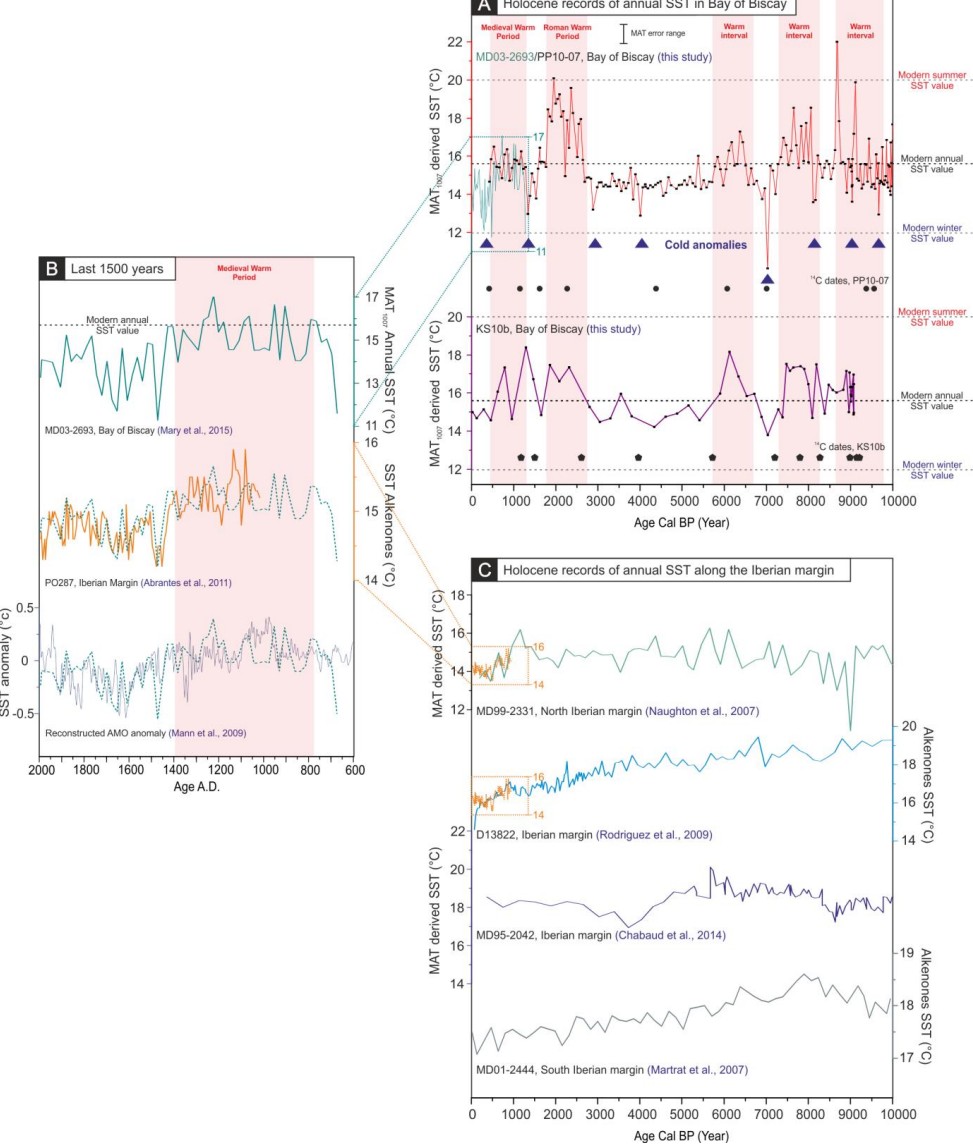

658
659

**Figure 3:** Mean Annual sea surface temperature (SST) records from the Western European margin. A) Holocene SST signals from cores PP10-07 and KS10b (this study) reconstructed using the Modern Analog Technique (MAT) based on planktonic foraminifera (see Methods), and compared to SST signal of the adjacent core MD03-2693 (Mary et al., 2015). Black dots identify 14C age control points. B) SST signals spanning the last 1500 years in the Bay of Biscay (core MD03-2693) based on MAT and from the Iberian Margin (core PO287-06, Abrantes et al., 2011) using alkenones. Reconstructed signals are compared with the AMO reconstruction of Mann et al., (2009). The dotted curve represents core MD03-2693 signal transposed on top of the two other curves. C) Holocene SST signals from the Iberian Margin using MAT based on planktonic foraminifera for cores MD99-2331 (after Naughton et al., 2007b) and MD95-2042 (after Chabaud et al., 2014) and Alkenones for cores D13882 (after Rodrigues et al., 2009) and MD01-2444 (after Martrat et al., 2007).


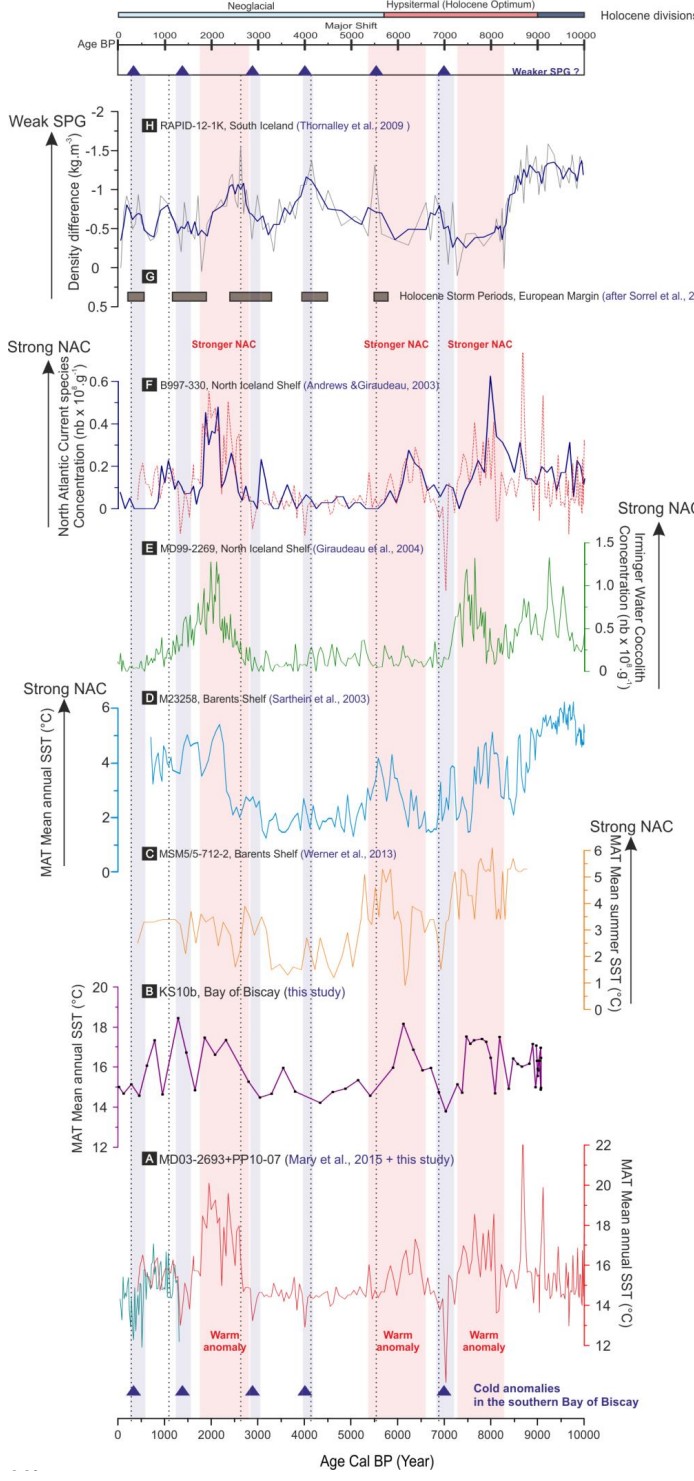

**Figure 4:** Comparison of annual SST Holocene signals from the Bay of Biscay (A and B) with records from the northern North Atlantic highlighting variations of the NAC intensity and SPG dynamics; C) SST signal of core MSM5/5-712-2 (Fram strait, Werner et al., 2013) and of D) core M23258 (Barents shelf, after Sarnthein et al., 2003), both reconstructed using the Modern Analog Technique (MAT) based on planktonic foraminifera; E) Concentration of NAC indicator coccolith species in core MD99-2269 (North Iceland Shelf, after Giraudeau et al., 2004) and in F) core B997-330 (North Iceland Shelf, after Andrews and Giraudeau, 2003); The PP10-07 record is here also plotted by a thin dotted red line to underline the comparison; G) Holocene Storm Periods (after Sorrel et al., 2012) reconstructed from sedimentological evidence from a compilation of coastal cores in North-western Europe; H) core Rapid-12-1K (Thornalley et al., 2009) proxy for upper-water column stratification, calculated using derived Mg/Ca and $\delta^{18}$O temperatures and salinities of *G. bulloides and G. inflata*. Dotted vertical lines point out events of density anomalies at sub-thermocline depths in the southern Iceland basin.

Changes in gyre circulation dynamics are compared with the Holocene division of Wanner et al. (2008). The topmost arrows indicate periods of probable weak SPG also corresponding to cold anomalies in the southern Bay of Biscay. Pink bands conversely highlight periods of warmth which also correspond to enhanced NAC activity North of Iceland.