# Peer review of "Changes in Holocene meridional circulation and poleward Atlantic 1 flow: the Bay of Biscay as a nodal point 2"

_Climate of the Past, 2016_

## Referee Comment (RC1) · Anonymous Referee #1 · 27 Apr 2016

Mary et al present an excellent new Holocene SST data set from the Bay of Biscay, including a very high resolution last 1500 years. Good reproducibility is shown between cores and at existing study sites off the Iberian margin, and many of the signals are seen in existing work further north, into the Nordic Seas. The figures are clearly presented. There are numerous instances where the language of the text could be improved, since the meaning is either unclear or very oddly worded, however I trust copy-editing will pick these up. Overall, the methods and results are very good, yet the discussion and interpretation could be improved.

My main criticism is that the authors often need to be more specific about precisely what the inferred mechanisms driving the changes are, and what their new insight is.

[Figure]

The authors draw attention to key findings in the conclusion, but not in the abstract. They interpret their data, alongside existing datasets, as showing regional differences (subpolar versus subtropical) in the timing and trends of temperature trends, notably between Iberian Margin data (subtropical), and the Bay of Biscay and North Iceland (subpolar). More specific and clearly worded conclusions regarding the drivers of these trends would be useful. The abstract needs improving by including specific key findings/results and interpretations. What is the specific important take home message and why is it important? Be precise.

Discussion of the results and inferred mechanistic scenarios are sometimes rather general ("a gyre-specific expression of the AMOC"). Can the authors go further than simply stating there are some regional differences across the North Atlantic (which has been demonstrated by numerous authors over the years (eg Moros et al 2006, PaleO; Solignac et al 2006, PaleO; deVernal and Hillaire-Marcel 2006, GPC; Thornalley et al 2009, Nature; Giraudeau et al 2010, QSR)? And perhaps of more importance, the addition of a discussion into why there is such good coherence between surface SST records between the Bay of Biscay and the North Iceland shelf, yet quite different trends to the sub-seasonal thermocline data south of Iceland (see comment for L198-200 below). Given that very different trends are observed between the surface and sub-surface south of Iceland, it seems likely the answer lies in different controls on surface versus sub-surface changes, as discussed by Thornalley et al 2009 - the subsurface being controlled by SPG dynamics whereas the surface being controlled by other factors. L169-L189 describes these surface changes, including two striking warm intervals, yet there is little discussion about the cause of these events, which are not seen in the subsurface records which are presumably monitoring SPG dynamics. And why is there a good match between the Bay of Biscay SST and the chosen North Iceland Shelf data, but not with numerous other records monitoring the eastern inflow of Atlantic water to the Nordic Seas (see comments for L83-188 below)?

This manuscript could be greatly improved with a little bit more thought and time spent

on drawing out the main mechanistic ideas and how they integrate with broader concepts and existing datasets of North Atlantic Holocene change – trying to be as precise as possible. I strongly encourage the authors to take such efforts since they have a very nice dataset to add to this debate, however, I would find it acceptable if it were published with only minor to its present form, since it does not, in my opinion, have any major factual inaccuracies and does an adequate (albeit limited) job of placing this dataset in context with some existing studies.

More specific comments:

L24: Is the Bay of Biscay a nodal position? How so? Often frontal shifts are envisaged shifting about a modal position of Newfoundland...

L30: I question whether this study actually offers unprecedented resolution (I would remove). Perhaps unprecedented for Bay of Biscay, but certainly not for the North Atlantic

Abstract: More generally this should also include a summary of the key findings, rather than just a brief description of the study site and methods.

L37: I find the implication that the AMOC controls the 'frequency' of climate over Europe confusing - what do you mean specifically (and cite ref.)

L46-48: This sentence uses a lot of jargon to say very little.

L47-49: the relevance of this sentence to the study is not that obvious.

L56: why is the Bay of Biscay ideally located? One could argue that sites further NW are closer to the STG/SPG boundary and so more sensitive to monitoring these changes.

L139: provide reference for support

L183-188: There are of course numerous other SST records available from the Nordic Seas under the path of the Inflow and NwAC (eg Risebrobakken et al 2003, Giraudeau

et al 2010, Rasmussen and Thomsen 2010) that have not been shown, many of which do not show similar patterns to the Bay of Biscay SST data. It would be interesting to think more about these different records, and more specifically why the Irminger Current/North Iceland shelf shows similar trends to the Bay of Biscay, but not the Faroe branch of the NAC (or at least a more mixed signal is seen in the NwAC and Barents Shelf), especially since one might initially expect a more direct link between the eastern limb extension of the NAC and the eastern located Bay of Biscay.

L198-200: This is incorrectly worded; more care is needed. The density anomalies in Thornalley et al are a combination of changes due to SPG driven changes in the seasonal sub-thermocline, and other changes in the surface water. Changes in the G. inflata record alone were interpreted as a SPG strength proxy, not the density difference, as plotted by Mary et al. Perhaps a case could be made that by taking the difference between the surface and the sub-thermocline layer removes any larger scale changes in SST and SSS, and helps isolate the SPG strength signal, although this would be at odds with Fig 3 in Thornalley et al 2009.

L203: The assertion that changes in density anomalies reported by Thornalley et al 2009 are synchronous with cold spells in Mary et al's record is unconvincing. Major features are sometimes in phase or out of phase. (The match between periods of storm activity and the SST data of this study is also not that striking.) This is not a major weakness in the paper, and perhaps it simply reflects that the Bay of Biscay SST is only weakly sensitive to expansion/contraction of the subpolar gyre, and at times these signals are dominated/swamped by other controls (perhaps of a more local origin, or of subtropical origin). Or the surface temperature records are less sensitive for monitoring changes in subpolar gyre dynamics than deeper thermocline records. Perhaps it would be worth adding such a caveat, rather than stretching the data comparison too far and inferring close relationships when they don't seem convincing. Yet the similarity between the Bay of Biscay SST and the North Iceland Shelf records is good. The question is therefore how to explain the coherence between the Bay of Biscay and

North Iceland SST records, and the different trends seen in the sub-seasonal thermocline data south of Iceland. Given similar differences are seen between the surface and sub-seasonal thermocline records at the same site south of Iceland (and if anything, the surface temperature data at this site looks more like the Bay of Biscay and North Iceland SST data – albeit not the similar!), rather than the explanation being found in regional differences, it is perhaps likely that it is to do with surface versus subsurface changes.

L210: please explain this inferred atmosphere-ocean interaction - be more specific.

L294: 'a decoupling of subpolar gyre dynamics' from what? This is unclear.

L300: please use alternative phrase to 'gyre-specific expression' – in essence you mean there are differing SST changes and trends in the subtropical and subpolar regions (or at least at the sites you discuss).

L312: unclear. What is meant by 'contrasted patterns'?

Technical corrections: L23: add 'in the subpolar North Atlantic to the end of first sentence' L34-35: remove this sentence - it adds nothing, and just reads oddly L49: 'rightly' should be 'correctly' L95: replace 'onwards' with 'using a' L173: please refer to figure panel this relates to L191: replace '-1oc' with '1oC cooler', otherwise it might be misread as if the temperature was -1oC! L283: replace 'extensions' with 'expansions' L605: add labels for what blue triangles are to figure caption L630: the plot is the density difference between the near-surface and base of the seasonal thermocline, not density anomalies at sub-thermocline depths as written in caption.

---

## Short Comment (SC1) · 30 Apr 2016

An excellent paper which fills important data gaps. Thanks for this new study. I have two minor points that I would like to raise:

1) In the text you cite Mojtahid et al. (2013) which however is not in the reference list and needs to be added.

2) You describe very interesting Holocene millennial-scale cycles. A typical data set from the North Atlantic against which such cycles are usually compared is from Bond et al. 2001. The Bond cycles were demonstrated to be solar-driven. http://science.sciencemag.org/content/294/5549/2130 It would be great if this comparison could be added to the paper.

---

## Author Comment (AC1) · 2 May 2016

We really appreciate the constructive comments of Reviewer # 1 and the very stimulating questions he raised. We will consider his/her remarks and suggestions "to boost" our manuscript for the final revision steps. We apologize for the language and the clumsy phrasing of some sections, and agree that our text is sometimes probably too vague and diluted, we want however to underline that it was thoughtfully revised by a native speaker. We will rewrite our manuscript in a more direct and persuasive style. We also completely agree with Reviewer # 1 suggestions on our discussion, i.e. that we could have tried to go a step forward, but were cautious in this version to avoid over-interpretations of the data. Considering Reviewer # 1 advices, we will go beyond the

simple observations and comparisons initially done, and will thus include an additional section and a Figure with a conceptual scheme, gathering and reconciling (if possible) the mechanistic functioning of the STP and SPG during the considered period.

---

## Short Comment (SC2) · 4 May 2016

We are very grateful for your positive comments and interesting suggestions.

The reference to the article of Mojtahid et al., (2013) is actually mentioned in the reference list (L454), although not at the correct position. This will be corrected in the final version. We apologize for the mistake and thank you for spotting it.

Regarding the influence of solar forcing, short-lived cold spells recorded in the SST signal of core PP10-07 at 8.2, 7, 4, 2.9 and 1.7 ka BP indeed show similarity with the so-called "Bond cycles", at ca 8, 6, 4.5, 3, 1.8 and 0.5ky (Bond et al., 2001). However, the very short duration of these events in the Bay of Biscay calls for a derived phe-

nomenon rather than a direct influence of solar forcing on SST oscillations. The same idea is indirectly suggested in our paper when we refer to the millennial-scale storminess maxima reconstruction (Figure n°3 in the manuscript) of Sorrel et al., (2012). These authors concluded that the solar forcing was not a primary trigger for storminess maxima but did not exclude its possible influence as a weak external driver.

Though, comparing the SST signal of PP10-07 core with Bond cycle proxies, such as drifted ice indices, or directly with solar irradiance signal is a challenging suggestion. We will definitively try such approach (see the preliminary Figure R1) and include it in the final revised version, if possible. For information, this comparison was done and discussed at the scale of the last 2 ka BP in our 2015 paper (Mary et al., 2015).

Moreover, Morley et al., (2014) suggest that the strength of the Latitudinal Thermal Gradient (LTG), driven by contrasting distribution of insolation between polar and tropical latitudes, impacts meridional heat transport by oceanic systems and associated teleconnections. A sharp increase of the LTG occurs around 2000 BP. Such forcing may enhance NAC inflow toward northern latitude, which may explain the large, multi-millennial scale anomalies visible on the Bay of Biscay.

References:

Bond, G., Kromer, B., Beer, J., Muscheler, R., Evans, M.N., Showers, W., Hoffmann, S., Lotti-Bond, R., Hajdas, I., Bonani, G. (2001): Persistent solar influence on North Atlantic climate during the Holocene. Science 294, 2130–2136.

Mary, Y., Eynaud, F., Zaragosi, S., Malaizé, B., Cremer, M. and Schmidt, S. (2015): High frequency environmental changes and deposition processes in a 2 kyr-long sedimentological record from the Cap-Breton canyon (Bay of Biscay), The Holocene, 25, 348–365, doi:10.1177/0959683614558647.

Mojtahid, M., Jorissen, F.J., Garcia, J., Schiebel, R., Michel, E., Eynaud, F., Gillet, H., Cremer, M., Diz Ferreiro, P., Siccha, M., Howa, H. (2013). High resolution Holocene record in the southeastern Bay of Biscay: Global versus regional climate signals. Palaeogeography, Palaeoclimatology, Palaeoecology 377, 28–44. doi:10.1016/j.palaeo.2013.03.004

Morley, A., Rosenthal, Y., deMenocal, P. (2014): Ocean-atmosphere climate shift during the mid-to-late Holocene transition. Earth and Planetary Science Letters 388, 18–26. doi:10.1016/j.epsl.2013.11.039

Roth, R. and Joos, F. (2013): A reconstruction of radiocarbon production and total solar irradiance from the Holocene 14C and CO2 records: implications of data and model uncertainties, Clim. Past, 9, 1879-1909, doi:10.5194/cp-9-1879-2013. data available from CP at http://www.clim-past.net/9/1879/2013/

Sorrel, P., Debret, M., Billeaud I., Jaccard S.L., McManus J.F., Tessier B. (2012): Persistent non-solar forcing of Holocene storm dynamics in coastal sedimentary archives, Nature Geoscience 12, 892–896. doi:10.1038/ngeo1619, 2012.
* * *
[Figure]

**Fig. 1.** Comparison of SST data of the Bay of Biscay with reconstructed solar irradiance data (after Roth and Joos, 2013) and reconstruction of Holocene storm periods (after Sorrel et al., 2012).

---

## Referee Comment (RC2) · J. Scourse (Referee) · 17 Aug 2016

Mary et al. present an excellent high resolution record of Holocene palaeoceanographic changes (SST) from the southern Bay of Biscay based on two closely-positioned cores. The SST record is based on MAT transfer functions on planktonic foraminiferal assemblages and is compared with other palaeoceanographic records from the Biscay/Iberian margin and the wider North Atlantic. The raw planktonic foram dataset is excellent and the way in which the transfer function has been applied is well explained. The data for the Roman Warm Period interval and their correlation with the wider North Atlantic datasets for this period are impressive. Records of this quality covering the entire Holocene are not common and it is important that the data are published.

However, 1. in places I feel there is a tendency to over-interpret the record, 2. sometimes the explanation is not as clear as it might be, 3. some fundamental contextual information is lacking, 4. independent lines of evidence to corroborate the transfer function SST reconstruction are lacking, and 5. most importantly, there are some generalised statements not supported by either numerical model simulations or tests of statistical significance.

At the outset (and in the Abstract) the authors emphasize the strategic location of the core sites in the context of the wider North Atlantic circulation/AMOC. It would be good to support this assertion with some spatial correlation plots between this site and wider North Atlantic SST/SSS fields over the calibration period. What key elements of the surface circulation correlate with SSTs at this location? The core locations are actually quite distal from the main centres of North Atlantic hydrographic variability so firming up this relationship with evidence is important. There is significant discussion in the Introduction on the relationships between the regional hydrography and the wider North Atlantic circulation, and with modes of North Atlantic climate variability (AMO/NAO) but this remains (and feels) speculative unless it can be supported by evidence. In terms of the excellent reconstructed time-series for the last 2000 years, how do these compare with the CMIP5 simulations, and the earlier data with the CMIP5 mid-Holocene simulations?

Whilst the quality/resolution of the foram-based transfer function SSTs are not in question, I would have liked to see some corroboration from independent data (e.g. oxygen isotopes, trace element ratios, alkenones) of at least sections of the record. The PP10-07 long Holocene record is spliced with data for the last 2000 years from MD03-2693; what are the correlation statistics for this overlap?

It is essential to provide some key information about the cores at the start of the Methods section. I note that the water depths are included in Table 1, but what is the geomorphological context of the core locations, why does the sedimentation rate differ so much between the two cores, what are the sediment sources to these locations including biogenic/lithic ratios and, in particular, what is the local hydrographic regime at this location and how does it relate to the wider North Atlantic circulation discussed above? It is also essential at this point to present lithostratigraphic logs for the cores. Unless these data have been published elsewhere they should be included here, or in the Supplementary info.

Detailed comments:

Some small grammatical/word selection changes are suggested on the attached annotated pdf.

Line 30 (and elsewhere): the records are described as being of "unprecedented" resolution. This has to be more specific – unprecedented for this region, for the North Atlantic? There are certainly sediment-based records of comparable resolution elsewhere and this record does not compare with annual-banded records of SST (coral, bivalves).

Line 34: be more specific over the temporal frequency being referred to here.

Line 49: "latitudinal and/or longitudinal migrations": do you literally mean migrations or intensification/relaxation of gyre circulation?

Lines 53-54: "this paper aims at testing Western European temperate oceanic signals vs. those from a broader North Atlantic view with a focus on the SPG dynamics": what is meant by "Western European temperate oceanic signals and how are these separated from broader North Atlantic/SPG dynamics. This seems a bit vague/loose to me.

Lines 94-96: this sentence requires rephrasing.

Line 128: what is an "undated point" in a surface sample dataset?

Line 156: what is meant by "focused" in this context?

Line 171: what do you mean by "typical"?

Line 187: what do you mean by the "modulation of the split" between the SPG and STG?

Lines 321-323: this is the last line of the Conclusion and I'm not clear what it actually means.

Figure 1: "seasonal" spelling in figure legend.

James Scourse Menai Bridge 17th August 2016

Please also note the supplement to this comment:
http://www.clim-past-discuss.net/cp-2016-32/cp-2016-32-RC2-supplement.pdf

———————————————————

[Figure]

**Supplement:**

[revised manuscript text omitted]

---

## Author Comment (AC2) · 8 Dec 2016

**CPD** Interactive discussion **on "Changes in Holocene meridional circulation and poleward Atlantic flow: the Bay of Biscay as a nodal point" by Yannick Mary et al.,**

We acknowledge the positive review of James Scourse on our paper and thank him for all the good suggestions and perspectives his review brings to our study.
All the comments done have been considered for the revision but as some details and complements will not be directly included in the revised manuscript, we have listed below the significant elements which sustain our results and findings, and could help readers to appreciate these interactive discussion and its topics at the best.

**CPD** Interactive comment **on "Changes in Holocene meridional circulation and poleward Atlantic flow: the Bay of Biscay as a nodal point" by Yannick Mary et al., by J. Scourse (Referee)**

**Topic of the discussion: general comments**
JS: Mary et al. present an excellent high resolution record of Holocene palaeoceanographic changes (SST) from the southern Bay of Biscay based on two closely positioned cores. The SST record is based on MAT transfer functions on planktonic foraminiferal assemblages and is compared with other palaeoceanographic records from the Biscay/Iberian margin and the wider North Atlantic. The raw planktonic foram dataset is excellent and the way in which the transfer function has been applied is well explained. The data for the Roman Warm Period interval and their correlation with the wider North Atlantic datasets for this period are impressive. Records of this quality covering the entire Holocene are not common and it is important that the data are published. **However,**
**1. in places I feel there is a tendency to over-interpret the record,**
**2. Sometimes the explanation is not as clear as it might be,**
**3. some fundamental contextual information is lacking,**
**4. independent lines of evidence to corroborate the transfer function SST reconstruction are lacking, and**
**5. most importantly, there are some generalized statements not supported by either numerical model simulations or tests of statistical significance.**

**Reply**: all these flaws are now corrected based on the whole review procedure. We have especially introduced new Figures and text sections to reinforce our observations and findings.

**Topic of the discussion: about the study site (hydrography and climatology)**
JS: At the outset (and in the Abstract) the authors emphasize **the strategic location of the core sites in the context of the wider North Atlantic circulation/AMOC**. It would be good to support this assertion with some spatial correlation plots between this site and wider North Atlantic SST/SSS fields over the calibration period. **What key elements of the surface circulation correlate with SSTs at this location?** The core locations are actually quite distal from the main centres of North Atlantic hydrographic variability so firming up this relationship with evidence is important. There is significant discussion in the Introduction on the relationships between the regional hydrography and the wider North Atlantic circulation, and with modes of North Atlantic climate variability (AMO/NAO) but this remains (and feels) speculative unless it can be supported by evidence.

**Reply**: many of the questionings legitimately raised by James Scourse about the context of the sites and the related forcings were in fact already considered in details in the Mary et al. (2015) paper - *ref: The Holocene Vol. 25(2) 348-365, DOI: 10.1177/0959683614558647*- which focusses on the MD03-2693 record. A very detailed description of the hydrological and sedimentological contexts is published in this article and we wanted to avoid repetitions in this new paper; we have however documented the modern hydrography and its modulation in the present article as a summary based on modern oceanographer's works (see line 63 to 88). The work of Garcia-Soto, Pingree et al. over the last 15 years are especially worth to consider as tests against the dominant

climatic modes were done regarding regional SST data ("Navidad structure and timing", e.g. Garcia-Soto et al. 2002).

**To reinforce the idea of the Bay of Biscay strategic location, an additional Figure has been included within Figure1 (i.e. Figure 1C below) and the citation of Mary et al. 2015 synthesis introduced when needed in the text.**

[Figure]

*New Figure 1C: SST evolution over the last centuries in the Bay of Biscay (from the MD03-2693 sedimentological record and from the compilation of Garcia-Soto et al., 2002) and comparison with the Global SST anomaly (after Kennedy, 2014), the Atlantic Tropical Cyclone Counts (after Landsea et al. 2010) and the NAO index of Hurell (http://research.jisao.washington.edu/data_sets/nao/).*

This new Figure 1C is built upon the last centuries and thus compiles the modern contextual hydrography and climate trends. To further test our reconstructions (even of very low resolution at this time scale, done on core MD03-2693, see Mary et al., 2015 for further elements), a 2°C shift in the modern annual SST mean (5 year running average after Garcia-Soto et al., 2002) has been applied for stressing the comparison with our study area. This value of 2 °C is justified by the southern and confined position of our sites within the Bay of Biscay which register the warmest oceanic conditions at this latitude in the North Atlantic (see especially the decadal average registered for summer months at http://www.nodc.noaa.gov/cgi-bin/OC5/woa13fv2/woa13fv2.pl?parameter=t, extracted images below).

[Figure]

We tested also this shift with the WOA sample tool (Table below, http://www.geo.uni-bremen.de/cgi-bin/woasample.pl):

| | LONG | LAT | Modern mean **Annual** SST (°C) | Modern mean **JFM** SST (°C) | Modern mean AMJ SST (°C) | Modern mean **JAS** SST (°C) | Modern mean OND SST (°C) | Nb point | **SST Seasonality modern range (°C)** |
|---|---|---|---|---|---|---|---|---|---|
| PP10-07 | -2.23 | 43.68 | **15.63** | **11.95** | 15.09 | **20.08** | 15.42 | 1 | **8.135** |
| Celtic margin area | - 4 | 50 | **12.322** | **9.705** | 11.078 | **15.369** | 13.135 | 2 | **5.66** |
| FROM: http://www.geo.uni-bremen.de/cgi-bin/woasample.pl, last consult 05/12/2016 | | | | | | | | | |

This compilation shows that (as already stated by physical oceanographers) a poor link exist with the NAO, even if modulations in SST oscillations seem to be coherent from a region to another. Not added on this Figure, but tested also, is the link with the Atlantic Multidecadal Oscillation (AMO) which, as stated by Garcia-Soto & Pingree (2012) is not straightforward but, probably, the most coherent driver of SST changes in the area.

**New Citations introduced:**

Kennedy, J.J., 2014. A review of uncertainty in in situ measurements and data sets of sea surface temperature: IN SITU SST UNCERTAINTY. Reviews of Geophysics 52, 1–32. doi:10.1002/2013RG000434

Landsea, C.W., Vecchi, G.A., Bengtsson, L., Knutson, T.R., 2010. Impact of Duration Thresholds on Atlantic Tropical Cyclone Counts*. Journal of Climate 23, 2508–2519.

**Topic of the discussion: about the reconstructions**
JS: In terms of the excellent reconstructed time-series for the last 2000 years, how do these compare with the CMIP5 simulations, and the earlier data with the CMIP5 mid-Holocene simulations?

**Reply**: This very stimulating question is not trivial as our reconstructed data are regional sea-surface temperatures and none of the products released up to now by CMIP5 (or the related PMIP3 experiment) are thus directly comparable. However, this is also one of the targets of the French ANR HAMOC project – see http://hamoc-interne.epoc.u-bordeaux1.fr/doku.php?id=start&#news - and works are thus going on within the involved French teams. This was not possible at this step to include a model-data comparison.

JS: Whilst the quality/resolution of the foram-based transfer function SSTs are not in question, **I would have liked to see some corroboration from independent data (e.g. oxygen isotopes, trace element ratios, alkenones) of at least sections of the record.**

**Reply**: Additional independent data (here XRF elemental ratio) have been added on the new Figure 5 (built to support mechanistical interpretations as required by Rev.1). In addition, on the illustration below, are compiled key XRF data (some will be included in the revised version) compared to those of foraminifera counts (relative abundance and SST) to highlight the sensitivity of this last tracer.

[Figure]

JS: The PP10-07 long Holocene record is spliced with data for the last 2000 years from MD03-2693; what are the correlation statistics for this overlap?

**Reply**: The overlap is absolutely not "forced": we have not changed anything in the age models to fit or tie our records. The statistics of the recovery are thus poor but that was not the purpose of the present work. For sure, with additional data and datings a composite record could be built since the coherency of the reconstructed SST is very strong and within the error bar (see the Table and Figures below where core to core results are compared within a time precision < 10 years).

**Zooming on the overlap:**

| Age (CE) | PP10-07 overlap-close ages | Age (CE) | MD03-2693 overlap-close ages | Delta |
|---|---|---|---|---|
| 675 | 15.5 | 671 | 12.2 | 3.2 |
| 727 | 15.4 | 736 | 15.5 | 0.1 |
| 780 | 16.3 | 788 | 16.1 | 0.2 |
| 833 | 15.6 | 834 | 14.4 | 1.2 |
| 887 | 15.8 | 904 | 16.6 | 0.8 |
| 942 | 15.9 | 951 | 16.7 | 0.8 |
| 997 | 15.4 | 997 | 14.9 | 0.5 |
| 1054 | 14.7 | 1065 | 16.3 | 1.5 |
| 1111 | 16.4 | 1110 | 15.2 | 1.2 |
| 1169 | 16.1 | 1178 | 15.9 | 0.2 |
| 1228 | 14.9 | 1223 | 17.1 | 2.2 |
| 1288 | 15.4 | 1291 | 14.8 | 0.6 |
| 1349 | 15.5 | 1341 | 16.0 | 0.6 |
| 1411 | 16.5 | 1404 | 15.8 | 0.7 |
| **1474** | **15.9** | **1472** | **11.9** | **4.0** |
| **1519** | **14.7** | **1517** | **15.0** | **0.3** |

[Figure]

[Figure]

JS: It is essential to provide some key information about the cores at the start of the Methods section. I note that the water depths are included in Table 1, but what is the geomorphological context of the core locations, why does the sedimentation rate differ so much between the two cores, what are the sediment sources to these locations including biogenic/lithic ratios and, in particular, what is the local hydrographic regime at this location and how does it relate to the wider North Atlantic circulation discussed above? It is also essential at this point to present lithostratigraphic logs for the cores. **Unless these data have been published elsewhere they should be included here, or in the Supplementary info.**

**Reply**: All the geomorphological and sedimentological contexts, as well as the lithostratigraphic descriptions of the cores have already been provided in details in the following references (cited in our article):

> Gaudin, M, Mulder, T., Cirac, P., Berne, S., and Imbert P.: Past and present sedimentation activity in the Capbreton Canyon, southern Bay of Biscay, Geo-Marine Letters 26, 331–345, 2006.
>
> Brocheray, S., Cremer, M., Zaragosi, S., Schmidt, S., Eynaud, F., Rossignol L., and Gillet, H.: 2000 years of frequent turbidite activity in the Capbreton Canyon (Bay of Biscay), Marine Geology, 347, 136–152, doi:10.1016/j.margeo.2013.11.009, 2014.
>
> Mojtahid, M., Jorissen, F.J., Garcia, J., Schiebel, R., Michel, E., Eynaud, F., Gillet, H., Cremer, M., Diz Ferreiro, P., Siccha, M., and Howa, H.: High resolution Holocene record in the southeastern Bay of Biscay: Global versus regional climate signals, Palaeogeography, Palaeoclimatology, Palaeoecology, 377, 28–44. doi:10.1016/j.palaeo.2013.03.004, 2013.
>
> Mary, Y., Eynaud, F., Zaragosi, S., Malaizé, B., Cremer, M. and Schmidt, S.: High frequency environmental changes and deposition processes in a 2 kyr-long sedimentological record from the Cap-Breton canyon (Bay of Biscay), The Holocene, 25, 348–365, doi:10.1177/0959683614558647, 2015.

Furthermore, calibration on modern planktonic foraminifera populations have been conducted within the following papers:

> Retailleau S., Eynaud F., Mary Y., Schiebel R., Howa H., 2012. An Ocean - Canyon head and river plume: how they may influence neritic planktonic foraminifera communities in the SE Bay of Biscay?, Journal of Foraminifera research 42(3), 257–269
>
> Retailleau S., Howa H., Schiebel R., Lombard F., Eynaud F., Schmidt S., Jorissen F., Labeyrie L., 2009. Planktic foraminiferal production along an offshore-onshore transect in the south-eastern Bay of Biscay. Continental Shelf research 29 (8), 1123-1135

These last references have been added in the text.

**Detailed comments:** all the detailed comments have been considered for the revision, but we just wanted to reply to one comment raised by both reviewers about the resolution of our record.

JS: Line 30 (and elsewhere): the records are described **as being of "unprecedented" resolution.** This has to be more specific – unprecedented for this region, for the North Atlantic? There are certainly sediment-based records of comparable resolution elsewhere and this record does not compare with annual-banded records of SST (coral, bivalves).

**Reply**: we agree that we have to be more specific but wanted to stress out that, up to now, none comparable SST record exists for a 10 ka long and continuous interval (with such a regular time slice). It is obvious on the Figure below where some records (those of the highest resolution have been selected and digitized from each related citations) were added for comparison on the basis of anonymous Rev1's suggestions.

This is a consideration that we will introduce in our text with some graphical supports in the new Figure 5.

---

## Author Response (AR1)

**Editor Decision: Reconsider after major revisions** (19 Nov 2016) by Dr. Thorsten Kiefer
Comments to the Author:

Dear Dr. Mary and co-authors,

Your manuscript had received two very constructive reviews and a short comment, which are in good agreement with each other. They were enthusiastic about the quality of the dataset you have generated, but ask for more elaborate discussion of the mechanisms behind the observed changes, more linguistically accurate expression of thoughts, more background information about the sediment cores, and clearer provision of inferences and conclusions.

I encourage you to submit a revised manuscript version that addresses the reviewers' requests as appropriate. Of course, as Ref#2 mentions and you also rightly point out in your response, this should not lead to an overinterpretation of the evidence.
Thank you for responding to the review of Referee#1 and to the short comment. It seems, however, that you did not respond to Ref#2 yet. Also your responses to Ref#1 were relatively brief and general, not responding systematically to each referee item one by one. When submitting your revised manuscript, I would therefore ask you to please provide a description of the changes you made, or alternatively, a manuscript version with the (relevant) changes tracked. **Moreover, please respond to the comments of Ref#2 in the form of an Author Comment in the Interactive Discussion**.
Thank you for stating that the data will be made available on Pangaea. I encourage you to archive the data there now already (protected if you want) in order to have them ready for release together with publication.
I look forward to receiving your revised manuscript.
Yours sincerely

Thorsten Kiefer

Dear Editor,
Please find below our reply to suggestions and comments formulated by the two reviewers (and also to the comments from Sebastian Luening).
We integrated all the suggestions they provided to improve this new version and hope we satisfy to their review.
Note that an Author comments (in response to James Scourse review) has been posted on the 08 demeber (*- AC2: 'Author Comment to Ref#2', Frederique Eynaud, 08 Dec 2016* and  *cp-2016-32-supplement).*

Faithfully yours,
Frederique Eynaud (on behalf of my co-authors)

**CPD** Interactive comment **on "Changes in Holocene meridional circulation and poleward Atlantic flow: the Bay of Biscay as a nodal point" by Yannick Mary et al.**

**Anonymous Referee #1**

REPLY to Referee #1:
*(see also **CPD** Interactive discussion - **AC1: 'Reply and acknowledgments to Anonymous reviewer # 1',** Frederique Eynaud, 02 May 2016 Printer-friendly Version)*

We really appreciate the constructive comments of Reviewer # 1 and the very stimulating questions he raised.
We have considered his/her remarks and suggestions "to boost" our manuscript for the final revision steps. We apologize for the language and the clumsy phrasing of some sections, and agree that our text is sometimes probably too vague and diluted, we want however to underline that it was thoughtfully revised by a native speaker. We will rewrite key parts of our manuscript in a more direct and persuasive style.
We also completely agree with Reviewer # 1 suggestions on our discussion, i.e. that we could have tried to go a step forward, but were cautious in this version to avoid over-interpretations of the data. Considering Reviewer # 1 advices, we tried to go beyond the simple observations and comparisons initially done, and have thus included additional discussions and a Figure (Figure 6) with a conceptual scheme, gathering and reconciling (if possible) the mechanistic functioning of the SPG during the considered period.

Mary et al present an excellent new Holocene SST data set from the Bay of Biscay, including a very high resolution last 1500 years. Good reproducibility is shown between cores and at existing study sites off the Iberian margin, and many of the signals are seen in existing work further north, into the Nordic Seas. The figures are clearly presented. There are numerous instances where the language of the text could be improved, since the meaning is either unclear or very oddly worded, however I trust copy-editing will pick these up. Overall, the methods and results are very good, yet the discussion and interpretation could be improved.
**My main criticism is that the authors often need to be more specific about precisely what the inferred mechanisms driving the changes are, and what their new insight is.**

The authors draw attention to key findings in the conclusion, but not in the abstract.
They interpret their data, alongside existing datasets, as showing regional differences (subpolar versus subtropical) in the timing and trends of temperature trends, notably between Iberian Margin data (subtropical), and the Bay of Biscay and North Iceland (subpolar**). More specific and clearly worded conclusions regarding the drivers of these trends would be useful. The abstract needs improving by including specific key findings/ results and interpretations.** What is the specific important take home message and why is it important? Be precise.

⇨ **Reply**: The abstract and conclusion have been rewritten; see now the new version in the revised manuscript (new sentences/ phrasing in red –correction mode).

Discussion of the results and inferred mechanistic scenarios are sometimes rather general ("a gyre-specific expression of the AMOC"). Can the authors go further than simply stating there are some regional differences across the North Atlantic (which has been demonstrated by numerous authors over the years (eg Moros et al 2006, PaleO; Solignac et al 2006, PaleO; deVernal and Hillaire-Marcel 2006, GPC; Thornalley et al 2009, Nature; Giraudeau et al 2010, QSR)? And perhaps of more importance, the addition of a discussion into why there is such good coherence between surface SST records between the Bay of Biscay and the North Iceland shelf, yet quite different trends to the sub-seasonal thermocline data south of Iceland (see comment for L198-200 below). Given that very different trends are observed between the surface and sub-surface south of Iceland, it seems likely the answer lies in different controls on surface versus sub-surface changes, as discussed by Thornalley et al 2009 – the subsurface being controlled by SPG dynamics whereas the surface being controlled by other factors.

⇨ **Reply**: We tried to satisfy this specific remark done by Reviewer 1 by adding new Figures (Figure 5 and 6) compiling additional records and conceptual schemes. Further discussion linked to this new Figures have been introduced in the text (see lines 353 to 374 )

L169-L189 describes these surface changes, including two striking warm intervals, yet there is little discussion about the cause of these events, which are not seen in the subsurface records which are presumably monitoring SPG dynamics. And why is there a good match between the Bay of Biscay SST and the chosen North Iceland Shelf data, but not with numerous other records monitoring the eastern inflow of Atlantic water to the Nordic Seas (see comments for L83-188 below)?

⇨ **Reply**: Eastern records have been considered also before our selection, and all depict good resemblance with the Bay of Biscay, even if the pattern is not so clear all over the last 10 ka. As pointed also later by Rev.1, amplitudes are not always comparable and some phasing artefacts due to the age models and to the local context, i.e. important fresh-water and sea-ice advection in the subpolar eastern Atlantic, may bias the comparison. The most resolved ones have been included in the new Figure 5.

This manuscript **could be greatly improved with a little bit more thought and time spent on drawing out the main mechanistic ideas** and how they integrate with broader concepts and existing datasets of North Atlantic Holocene change – trying to be as precise as possible. I strongly encourage the authors to take such efforts since they have a very nice dataset to add to this debate, however, I would find it acceptable if it were published with only minor to its present form, since it does not, in my opinion, have any major factual inaccuracies and does an adequate (albeit limited) job of placing this dataset in context with some existing studies.

⇨ **Reply**: We followed Reviewer 1 advice and have added a conceptual scheme and a Table of bibliographic compilation that will be included in the supplementary information (as Table E2) to tentatively provide a mechanistic scenario. They are also presented below (see the legend in the revised manuscript).

[Figure]

BB Warm events,
large extension of the SPG

BB cool events,
contracted SPG

| Forcings/ mechanisms | Bond et al. 2001 | | Thornalley et al. 2009 | | Giraudeau et al. 2010 | Sorel et al. 2012 | | Staine-Urias et al. 2013 | | Morley et al. 2014 | | Synthesis on BB SST anomalies (This work) integrating the comparison with key Holocene sequences (Fig 4 and 5) | |
|---|---|---|---|---|---|---|---|---|---|---|---|---|---|
| **SPG strength**
 SPG extension | *not specified* | | "strong "
 longitudinal (E-W) ↔ | "weak"
 latitudinal (N-S) ↕ | *not specified* | longitudinal (E-W) ↔ | "weak"
 latitudinal (N-S) ↕ | "strong "
 longitudinal (E-W) ↔ | "weak"
 latitudinal (N-S) ↕ | "strong "
 longitudinal (E-W) ↔ | "weak"
 latitudinal (N-S) ↕ | "strong" if we follow the consensus but divergent pattern with IC
 longitudinal (E-W) ↔ | "weak" if we follow the consensus but divergent pattern with IC
 latitudinal (N-S) ↕ |
| NAO index | rather NAO- (but not a basin wide expression) | | | | NAO like pattern | NAO+ | NAO- | NAO+ | NAO- | modern conditions | 1930 condtions | NAO+ ? If based on the Medieval anomaly | |
| **Atlantic Inflow in the Nordic seas**
 IC/ Denmark strait pathway | | | | | low | high | | | *not specified* | *not specified* | low inflow | high inflow | high inflow detected at hight latitude of the GIN seas (Barents sea margin)
 high inflow except at 6 ka | lowest IC inflow |
| South iceland salinity | | | Saline intervals | | | | | | | | | | saline |
| South iceland upper water stratification | | | low (negative) density diff | | | | | | | | | Different patterns if late or early Holocene | rather low (negative) density diff |
| Westerlies/ Europe | | | decreasing wind stress | | | | shifted to the south | strong, warmth/ moist | | | | | |
| **Storms over Europe**
 Climate over Europe | | | | | | low activity | high activity | | cool events | | | **low activity**
 Warm | high activity
 Coolings |
| Freshening/ export of sea-ice along Greenland | low | high | increase | | | low | high | | | | | Different patterns if late or early Holocene (residual ice-sheet melting ?) | |
| Solar (nuclide production) | minima | maxima | | | | | | | | | | Not obvious see reply to Sebastian Luening' comment and Figure 5 | |

**More specific comments:**

L24: Is the Bay of Biscay a nodal position? How so? Often frontal shifts are envisaged shifting about a modal position of Newfoundland...

**Reply**: to reinforce this idea, we have added a zoom on the last centuries which monitor SST changes from different regions and their trends along this period (see also the reply to Rev 2 - First Figure).

L30: I question whether this study actually offers unprecedented resolution (I would remove). Perhaps unprecedented for Bay of Biscay, but certainly not for the North Atlantic

**Reply**: the term "unprecedented" has been changed (see also the reply to Rev 2 ).

Abstract: More generally this should also include a summary of the key findings, rather than just a brief description of the study site and methods.

**Reply**: done, see first reply above.

L37: I find the implication that the AMOC controls the 'frequency' of climate over Europe confusing - what do you mean specifically (and cite ref.)

**Reply**: we rephrased the sentence and introduced references which were deleted with the first sentence of this section (as recommended in Technical corrections of Rev.1).

L46-48: This sentence uses a lot of jargon to say very little.
L47-49: the relevance of this sentence to the study is not that obvious.

**Reply**: deleted.

L56: why is the Bay of Biscay ideally located? One could argue that sites further NW are closer to the STG/SPG boundary and so more sensitive to monitoring these changes.

**Reply**: we hope that the new insert (C) added in Figure 1 monitoring the last centuries will convince Rev 1 and the readers about the key location of the site. Further arguments are also added in the text line 70 to 77.

L139: provide reference for support
**Reply**: done. Line 158

L183-188: There are of course numerous other SST records available from the Nordic Seas under the path of the Inflow and NwAC (eg Risebrobakken et al 2003, Giraudeau et al 2010, Rasmussen and Thomsen 2010) that have not been shown, many of which do not show similar patterns to the Bay of Biscay SST data. It would be interesting to think more about these different records, and more specifically why the Irminger Current/North Iceland shelf shows similar trends to the Bay of Biscay, but not the Faroe branch of the NAC (or at least a more mixed signal is seen in the NwAC and Barents Shelf), especially since one might initially expect a more direct link between the eastern limb extension of the NAC and the eastern located Bay of Biscay.

**Reply**: These records were all considered before the selections. Actually, in the frame of the ANR HAMOC, we have built a database selecting high resolution records notably obtained on the basis on planktonic foraminifera (see http://hamoc-interne.epoc.u-bordeaux1.fr/doku.php?id=start&#news). The new Figure 5 compiles those presented in the Risebrobakken et al., study (MD95-2011 /MD99-2284) in 2013. (see also the reply to Rev 2 ).

L198-200: This is incorrectly worded; more care is needed. The density anomalies in Thornalley et al are a combination of changes due to SPG driven changes in the seasonal sub-thermocline, and other changes in the surface water. Changes in the G. inflate record alone were interpreted as a SPG strength proxy, not the density difference, as plotted by Mary et al. Perhaps a case could be made that by taking the difference between the surface and the sub-thermocline layer removes any larger scale changes in SST and SSS, and helps isolate the SPG strength signal, although this would be at odds with Fig 3 in Thornalley et al 2009.

**Reply**: we have possibly over-interpreted the Thornalley et al., 2009 record but our interpretation was based on their caption in Figure 2 (see below). We reproduced on Figure 4 the density difference curve.

[Figure]

**Figure 2 | Proxy records for RAPiD-12-1K. a, b,** Mg/Ca-based temperatures (**a**) and salinity estimates derived from paired Mg/Ca–δ¹⁸O measurements (**b**), for near-surface (*G. bulloides,* red) and sub-thermocline (*G. inflata,* blue) waters. Also shown is a scale bar for δ¹⁸O$_{sw}$ values, corrected for whole-ocean ice-volume changes. **c, d,** Proxies for upper-water-column stratification (stratification increases upwards) based on the δ¹⁸O difference between *G. bulloides* and *G. inflata* (**c**) and the inferred water density difference between *G. bulloides* and *G. inflata* (**d**), calculated using derived temperatures and salinities. Three-point running means shown in bold. LIA, Little Ice Age.

L203: The assertion that changes in density anomalies reported by Thornalley et al 2009 are synchronous with cold spells in Mary et al's record is unconvincing. Major features are sometimes in phase or out of phase. (The match between periods of storm activity and the SST data of this study is also not that striking.) This is not a major weakness in the paper, and perhaps it simply reflects that the Bay of Biscay SST is only weakly sensitive to expansion/contraction of the subpolar gyre, and at times these signals are dominated/swamped by other controls (perhaps of a more local origin, or of subtropical origin). Or the surface temperature records are less sensitive for monitoring changes in subpolar gyre dynamics than deeper thermocline records. Perhaps it would be worth adding such a caveat, rather than stretching the data comparison too far and inferring close relationships when they don't seem convincing. Yet the similarity between the Bay of Biscay SST and the North Iceland Shelf records is good.
**Reply**: The nuances in the peak to peak correlation have been added in our text line 221.

**The question is therefore how to explain the coherence between the Bay of Biscay and North Iceland SST records**, and the different trends seen in the sub-seasonal thermocline data south of Iceland. Given similar differences are seen between the surface and sub-seasonal thermocline records at the same site south of Iceland (and if anything, the surface temperature data at this site looks more like the Bay of Biscay and North Iceland SST data – albeit not the similar!), rather than the explanation being found in regional differences, it is perhaps likely that it is to do with surface versus subsurface changes.
**Reply**: we expanded our discussion but lack additional subsurface data in the Bay of Biscay to introduce such a debate without over-interpretation. This will be done in a further study as works are ongoing in the frame of the French ANR HAMOC.

L210: please explain this inferred atmosphere-ocean interaction - be more specific. Changed, see line 232

L294: 'a decoupling of subpolar gyre dynamics' from what? This is unclear. Changed, see additional discussion section line 328

L300: please use alternative phrase to 'gyre-specific expression' – in essence you mean there are differing SST changes and trends in the subtropical and subpolar regions (or at least at the sites you discuss). Changed, see line 326

L312: unclear. What is meant by 'contrasted patterns'? Changed, see 361

*Technical corrections:*
L23: add 'in the subpolar North Atlantic to the end of first sentence' Already in the original version but we deleted sub-polar

L34-35: remove this sentence - it adds nothing, and just reads oddly Removed
L49: 'rightly' should be 'correctly' The corresponding sentence has been deleted.
L95: replace 'onwards' with 'using a' Corrected
L173: please refer to figure panel this relates to Done
L191: replace'-1oc' with '1oC cooler', otherwise it might be misread as if the temperature was -1oC! Done
L283: replace 'extensions' with 'expansions' Done
L605: add labels for what blue triangles are to figure caption Done
L630: the plot is the density difference between the near-surface and base of the seasonal thermocline, not density anomalies at sub-thermocline depths as written in caption. Corrected
An excellent paper which fills important data gaps. Thanks for this new study. I have two minor points that I would like to raise:

REPLY to Sebastian Luening:
*(see also* **CPD** *Interactive discussion* **- SC2: 'Reply and Comments to Holocene Bond Cycle', Yannick Mary, 04 May 2016)**

We are very grateful for your positive comments and interesting suggestions, and have integrated your comments in the revisions.

1) In the text you cite Mojtahid et al. (2013) which however is not in the reference list and needs to be added.
**Reply**: The reference to the article of Mojtahid et al., (2013) is actually mentioned in the reference list (L454), although not at the correct position. This has been corrected in the final version. We apologize for the mistake and thank you for spotting it.

2) You describe very interesting Holocene millennial-scale cycles. A typical data set from the North Atlantic against which such cycles are usually compared is from Bond et al. 2001. The Bond cycles were demonstrated to be solar-driven. http://science.sciencemag.org/content/294/5549/2130 It would be great if this comparison could be added to the paper.
**Reply**: **The new Figure 5 integrates Bond et al., 2001 data together with the Roth & Joos, 2013 dataset.** Regarding the Solar forcing (as previously mentioned in the interactive discussion), short-lived cold spells recorded in the SST signal of core PP10-07 at 8.2, 7, 4, 2.9 and 1.7 ka BP indeed show similarity with the so-called "Bond cycles", at ca 8, 6, 4.5, 3, 1.8 and 0.5ky (Bond et al., 2001). However, the very short duration of these events in the Bay of Biscay calls for a derived phenomenon rather than a direct influence of solar forcing on SST oscillation. Though, comparing the SST signal of PP10-07 core with Bond cycle proxies, such as drifted ice indices, or directly with solar irradiance signal is a challenging suggestion we were tempted initially to do. This comparison is indirectly done in our paper thanks to the comparison with the Sorel et al. millennial-scale storminess maxima we refer to. These authors concluded that the solar forcing was not a primary trigger but did not excluded its possible influence as a weak external driver. For information, a similar comparison was done and discussed at the scale of the last 2 ka BP in our 2015 paper (Mary et al., 2015).
Moreover, Morley et al., (2014) suggest that the strength of the Latitudinal Thermal Gradient (LTG), driven by contrasting distribution of insolation between polar and tropical latitudes, impacts meridional heat transport by oceanic systems and associated teleconnections. A sharp increase of the LTG occurs around 2000 BP. Such forcing may enhance NAC inflow toward northern latitude, which may explain the large, multi-millennial scale anomalies visible on the Bay of Biscay.

| | LONG | LAT | Modern mean **Annual** SST (°C) | Modern mean **JFM** SST (°C) | Modern mean AMJ SST (°C) | Modern mean **JAS** SST (°C) | Modern mean OND SST (°C) | Nb point | SST Seasonality modern range (°C) |
|---|---|---|---|---|---|---|---|---|---|
| PP10-07 | -2.23 | 43.68 | **15.63** | **11.95** | 15.09 | **20.08** | 15.42 | 1 | **8.135** |
| Celtic margin area | - 4 | 50 | **12.322** | **9.705** | 11.078 | **15.369** | 13.135 | 2 | **5.66** |
| FROM: http://www.geo.uni-bremen.de/cgi-bin/woasample.pl, last consult 05/12/2016 | | | | | | | | | |

[Figure]

*New Figure 1C: SST evolution over the last centuries in the Bay of Biscay (from the MD03-2693 sedimentological record and from the compilation of Garcia-Soto et al., 2002) and comparison with the Global SST anomaly (after Kennedy, 2014), the Atlantic Tropical Cyclone Counts (after Landsea et al. 2010) and the NAO index of Hurell (http://research.jisao.washington.edu/data_sets/nao/).*

*REF: Kennedy, J.J., 2014. A review of uncertainty in in situ measurements and data sets of sea surface temperature: IN SITU SST UNCERTAINTY. Reviews of Geophysics 52, 1–32. doi:10.1002/2013RG000434*
*Landsea, C.W., Vecchi, G.A., Bengtsson, L., Knutson, T.R., 2010. Impact of Duration Thresholds on Atlantic Tropical Cyclone Counts\*. Journal of Climate 23, 2508–2519.*

This exercise (new compilation in insert FIGURE 1C) shows that (as already stated by physical oceanographers) a poor link exist with the NAO, even if modulations in SST oscillations seem to be coherent from a region to another. Not added on this Figure, but tested also, is the link with the Atlantic Multidecadal Oscillation (AMO) which, as stated by Garcia-Soto & Pingree (2012) is not straightforward but, probably, the most coherent driver of SST changes in the area.

JS: In terms of the excellent reconstructed time-series for the last 2000 years, how do these compare with the CMIP5 simulations, and the earlier data with the CMIP5 mid-Holocene simulations?

Reply: This very stimulating question is not trivial as our reconstructed data are regional sea-surface temperatures and none of the products released up to now by CMIP5 (or the related PMIP3 experiment) are thus directly comparable. However, this is also one of the targets of the French ANR HAMOC project – see http://hamoc-interne.epoc.u-bordeaux1.fr/doku.php?id=start&#news - and works are thus ongoing within the involved French teams. This was not possible at this step to include a model-data comparison.

JS: Whilst the quality/resolution of the foram-based transfer function SSTs are not in question, **I would have liked to see some corroboration from independent data (e.g. oxygen isotopes, trace element ratios, alkenones) of at least sections of the record.**

Reply: Additional independent data (here XRF elemental ratio measured in PP10-07) have been added on the new Figure 5 (built to support mechanistic interpretations as required by Rev.1): we thus added in the revised article: XRF ratio (Ca/Si, Rb/Sr the latter being a grain-size proxy) and planktonic foraminifera absolute abundances in PP10-07

JS: The PP10-07 long Holocene record is spliced with data for the last 2000 years from MD03-2693; what are the correlation statistics for this overlap?

[Figure]

Reply: The overlap is absolutely not "forced": we have not changed anything in the age models to fit or tie our records.
The statistics of the recovery are thus poor but that was not the purpose of the present work. For sure, with additional data and datings a composite record could be built since the coherency of the reconstructed SST is very strong and within the error bar (see the Table and Figures provided with the *- AC2: 'Author Comment to Ref#2', Frederique Eynaud, 08 Dec 2016 and its cp-2016-32-supplement*).

JS: It is essential to provide some key information about the cores at the start of the Methods section. I note that the water depths are included in Table 1, but what is the geomorphological context of the core locations, why does the sedimentation rate differ so much between the two cores, what are the sediment sources to these locations including biogenic/lithic ratios and, in particular, what is the local hydrographic regime at this location and how does it relate to the wider North Atlantic circulation discussed above? It is also essential at this point to present lithostratigraphic logs for the cores. **Unless these data have been published elsewhere they should be included here, or in the Supplementary info.**

Reply: All the geomorphological and sedimentological contexts, as well as the lithostratigraphic descriptions of the cores have already been provided in details in the following references (cited in our article):

Gaudin, M, Mulder, T., Cirac, P., Berne, S., and Imbert P.: Past and present sedimentation activity in the Capbreton Canyon, southern Bay of Biscay, Geo-Marine Letters 26, 331–345, 2006.

Brocheray, S., Cremer, M., Zaragosi, S., Schmidt, S., Eynaud, F., Rossignol L., and Gillet, H.: 2000 years of frequent turbidite activity in the Capbreton Canyon (Bay of Biscay), Marine Geology, 347, 136–152, doi:10.1016/j.margeo.2013.11.009, 2014.

Mojtahid, M., Jorissen, F.J., Garcia, J., Schiebel, R., Michel, E., Eynaud, F., Gillet, H., Cremer, M., Diz Ferreiro, P., Siccha, M., and Howa, H.: High resolution Holocene record in the southeastern Bay of Biscay: Global versus regional climate signals, Palaeogeography, Palaeoclimatology, Palaeoecology, 377, 28–44. doi:10.1016/j.palaeo.2013.03.004, 2013.

Mary, Y., Eynaud, F., Zaragosi, S., Malaizé, B., Cremer, M. and Schmidt, S.: High frequency environmental changes and deposition processes in a 2 kyr-long sedimentological record from the Cap-Breton canyon (Bay of Biscay), The Holocene, 25, 348–365, doi:10.1177/0959683614558647, 2015.

Furthermore, calibration on modern planktonic foraminifera populations have been conducted within the following papers:

Retailleau S., Eynaud F., Mary Y., Schiebel R., Howa H., 2012. An Ocean - Canyon head and river plume: how they may influence neritic planktonic foraminifera communities in the SE Bay of Biscay?, Journal of Foraminifera research 42(3), 257–269

Retailleau S., Howa H., Schiebel R., Lombard F., Eynaud F., Schmidt S., Jorissen F., Labeyrie L., 2009. Planktic foraminiferal production along an offshore-onshore transect in the south-eastern Bay of Biscay. Continental Shelf research 29 (8), 1123-1135

These last references have been added in the text (lines 171-172)

**Detailed comments:**

Some small grammatical/word selection changes are suggested on the attached annotated pdf.
**Reply**: all the detailed comments have been considered for the revision and we thank JS for his patient editing of our manuscript.

JS: Line 30 (and elsewhere): the records are described **as being of "unprecedented" resolution.** This has to be more specific – unprecedented for this region, for the North Atlantic? There are certainly sediment-based records of comparable resolution elsewhere and this record does not compare with annual-banded records of SST (coral, bivalves).
**Reply**: we agree that we have to be more specific but wanted to stress out that, up to now, none comparable SST record exists for a 10 ka long and continuous interval (with such a regular time slice). We have however modulated our sentence.

Line 34: be more specific over the temporal frequency being referred to here. **Reply**: this sentence was deleted (see Rev1 comments)

Line 49: "latitudinal and/or longitudinal migrations": do you literally mean migrations or intensification/relaxation of gyre circulation? **Reply**: sentence changed, see line 58/59

Lines 53-54: "this paper aims at testing Western European temperate oceanic signals vs. those from a broader North Atlantic view with a focus on the SPG dynamics": what is meant by "Western European temperate oceanic signals and how are these separated from broader North Atlantic/SPG dynamics. This seems a bit vague/loose to me.
**Reply**: changed by "at testing Western European temperate oceanic signals *vs*. those from a broader North Atlantic…" and simplified, see line 64

Lines 94-96: this sentence requires rephrasing. **Reply**: change by "Here we present past Holocene SST data reconstructed after an ecological transfer function based on the Modern Analogue Technique (see Methods) applied to planktonic foraminiferal assemblages." The following sentence has also been slightly modified. see line 111

Line 128: what is an "undated point" in a surface sample dataset? **Reply**: changed by "non-stratigraphically constrained"

Line 156: what is meant by "focused" in this context? **Reply**: changed by "studied"

Line 171: what do you mean by "typical"? **Reply**: changed by "characteristic"

Line 187: what do you mean by the "modulation of the split" between the SPG and STG? **Reply**: changed and simplified by "…and to its split between the SPG and the STG".

Lines 321-323: this is the last line of the Conclusion and I'm not clear what it actually means. **Reply**: reformulated

Figure 1: "seasonal" spelling in figure legend. ? **Reply**: corrected

[revised manuscript text omitted]

**Figure 1 (revised)**

[Figure]

                          Figure 2

[Figure]

     **Figure 3**

[Figure]

**Figure 4**

[Figure]

**Figure 5 (new Figure)**

[Figure]

**Figure 6 (new Figure)**